# RNA polymerase II is a polar roadblock to a progressing DNA fork

Taryn M. Kay [1], James T. Inman[2,3], Lucyna Lubkowska[4], Tung T. Le[2,3], Jin Qian[2], Porter M. Hall [2,3], Sahil Batra [5], Dirk Remus [5], Dong Wang [6,7,8], Mikhail Kashlev [4] & Michelle D. Wang [2,3] ✉

Transcription–replication conflicts threaten genome stability. Although head-on conflicts are more detrimental and prone to R-loop formation than co-directional conflicts, the cause of this RNA polymerase roadblock polarity remains unclear, and proposed structures of these R-loops are speculative. Here, we examine the Pol II roadblock to a DNA fork advanced by mechanical unzipping to mimic replisome progression. We found that a head-on Pol II with a minimal transcript resists disruption more strongly, revealing inherent polarity. Moreover, an elongating Pol II with a long RNA transcript becomes an even more potent roadblock, mediated by RNA–DNA hybrid formation. Surprisingly, when a Pol II collides with the DNA fork head-on and becomes backtracked, a hybrid can form in front of Pol II, creating a topological lock that traps Pol II at the fork. Our findings capture the basal properties of Pol II interactions with a DNA fork, revealing significant implications for transcription–replication conflicts.

Efficient and accurate DNA replication is essential for the preservation and transmission of the genetic information of all living organisms, from bacteria to humans. However, DNA replication shares the same DNA substrate with transcription, another essential cellular function that occurs throughout the cell cycle. Thus, a replisome may encounter a transcribing RNA polymerase (RNAP)[1–9], resulting in a conflict that could lead to significant cellular consequences. A replisome may encounter an RNAP moving either co-directionally or head-on, and the two orientations are known to lead to different outcomes concerning genome stability and integrity[1,10,11]. While both orientations are disruptive to replication, head-on conflicts are much more detrimental than co-directional conflicts[1,4,10,12–18]. When a replisome encounters an RNAP head-on, replication progression is greatly impeded, and the replisome can be stalled. This may lead to replisome disassembly, fork reversal, DNA double-strand breaks, and activation of the DNA damage response[9,10,19,20]. In contrast, while co-directional conflicts are still disruptive to replication, they do not stall the replisome to the same extent as head-on conflicts[2,10,20].

Despite the cellular consequences of these conflicts, our understanding of their underlying mechanisms remains limited. Although RNAP has been found to be a more severe roadblock to replication when approaching a replisome head-on vs co-directionally[1,4,10,12–18], the fundamental cause for the RNAP roadblock polarity remains unclear. Furthermore, head-on conflicts, not co-directional ones, have been found to promote the formation and accumulation of R-loops, three-stranded RNA–DNA hybrid structures with the nascent RNA transcript reannealed to the template DNA strand at the region of the encounter[21–23]. However, the structure of these R-loops remains largely speculative. The prevailing view generally places the R-loop behind the RNAP in the context of head-on replication–transcription conflicts, but this view

[1]Biophysics Program, Cornell University, Ithaca, NY, USA. [2]Department of Physics & LASSP, Cornell University, Ithaca, NY, USA. [3]Howard Hughes Medical Institute, Cornell University, Ithaca, NY, USA. [4]RNA Biology Laboratory, Center for Cancer Research, National Cancer Institute, Frederick, MD, USA. [5]Molecular Biology Program, Memorial Sloan Kettering Cancer Center, New York, NY, USA. [6]Division of Pharmaceutical Sciences, Skaggs School of Pharmacy and Pharmaceutical Sciences, University of California San Diego, La Jolla, CA, USA. [7]Department of Cellular & Molecular Medicine, University of California San Diego, La Jolla, CA, USA. [8]Department of Chemistry and Biochemistry, University of California San Diego, La Jolla, CA, USA. ✉e-mail: mwang@physics.cornell.edu

has not been validated experimentally[23–25]. Understanding the consequences of head-on conflicts requires a method that can elucidate the nature of these R-loops, which has proven experimentally challenging.

In this work, we have employed a model system involving a mechanically progressing DNA fork and an elongating Pol II. The progressing DNA fork used here somewhat resembles a progressing replication fork in that the DNA fork is mechanically unwound rather than being unwound by CMG. A complete understanding of RNA–DNA hybrid formation during a transcription–replication conflict requires the full reconstitution of both machineries, an experimental feat that has yet to be demonstrated biochemically with eukaryotic systems. Meanwhile, the simplicity of this approach makes it possible to directly investigate Pol II roadblock polarity and the structure of RNA–DNA hybrid. Our findings provide significant insights into the nature of the RNAP roadblock that are relevant to understanding transcription–replication conflicts.

## Results

### Pol II is an inherent polar roadblock to a progressing DNA fork

It remains unclear whether RNAP inherently acts as a polar roadblock to replication, or whether head-on conflicts promote R-loop formation, which in turn strengthens RNAP as a barrier to replication. To examine the RNAP roadblock polarity, we mimicked the replisome progression by mechanically unzipping DNA through a bound yeast Pol II elongation complex using a dual optical trap equipped with a multi-channel flow cell (Fig. 1a and Supplementary Fig. 1). Previously, we demonstrated that the DNA unzipping mapper is a powerful approach for mapping protein–DNA interactions by rapidly advancing the DNA fork through the DNA-bound protein to disrupt the interactions[26–30]. Here, we use the DNA unzipping mapper to investigate the resistance experienced by the DNA fork when Pol II transcribes either in the same direction (co-directional orientation) or in the opposite direction (head-on orientation) of an advancing fork.

In this experiment, we formed a Pol II elongation complex on a dsDNA template[31], which was ligated to two dsDNA unzipping adaptor

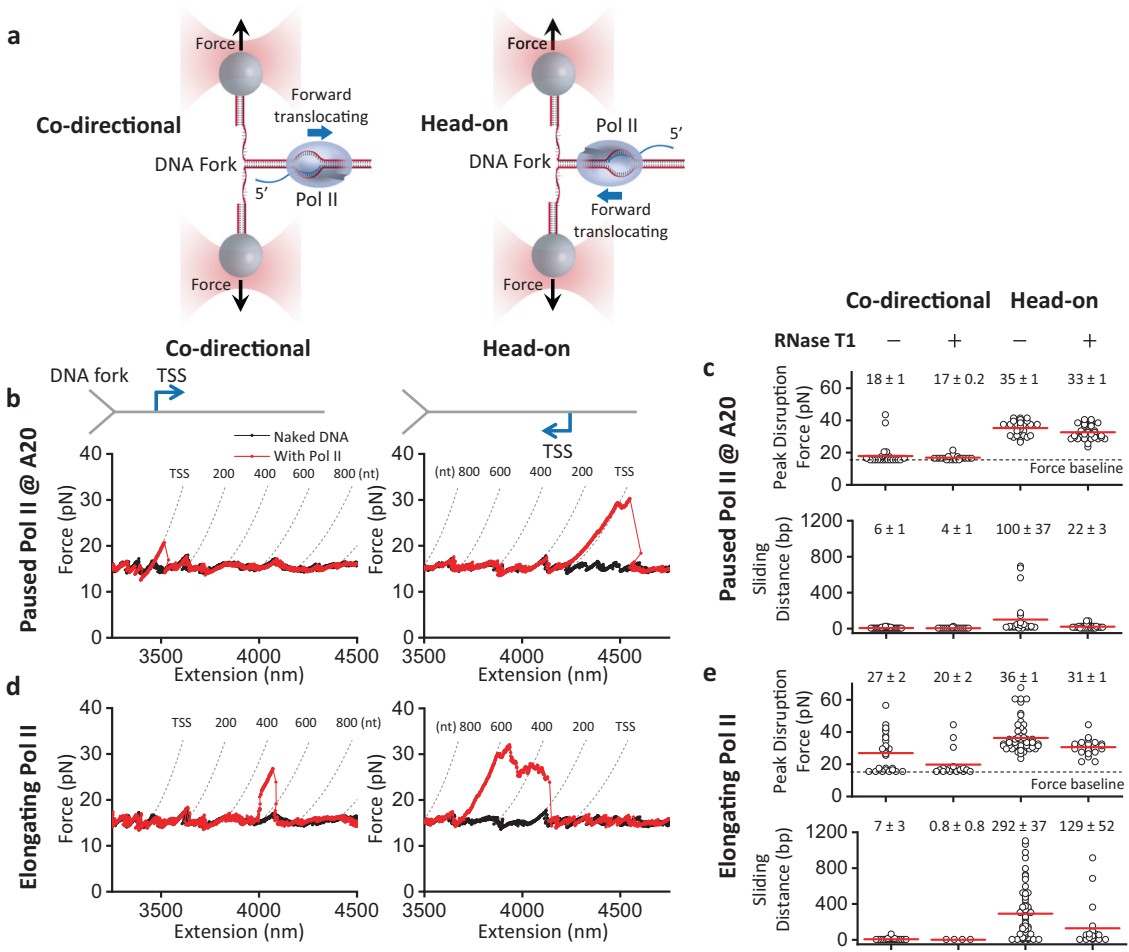

**Fig. 1 | Pol II is a polar roadblock to a progressing DNA fork. a** Experimental configuration showing co-directional (CD) and head-on (HO) collision orientations between a DNA fork and Pol II. The two daughter strands are tethered between two optically trapped beads. The DNA fork is mechanically unzipped through a Pol II elongation complex at a velocity of 100 nm/s, and force-extension data reveal the strength and location of the Pol II–DNA interactions. **b** Representative force-extension traces of the DNA fork unzipping through a paused Pol II at A20 in both orientations. Pol II is paused 20 nt from the transcription start site (TSS). Dashed curves show predicted force-extension profiles for forks encountering Pol II after transcription of the specified number of nucleotides. **c** Scatter plots of the peak disruption force and Pol II sliding distance for both orientations of paused Pol II without and with RNase T1. The mean values (also as the red bars) and their SEMs are indicated. Sample sizes: CD: $N = 38$ (−RNase T1), $N = 24$ (+RNase T1); HO: $N = 28$ (−RNase T1), $N = 39$ (+RNase T1). **d** Representative force-extension traces of the DNA fork unzipping through an elongating Pol II in both orientations. **e** Scatter plots of the peak disruption force and Pol II sliding distance for both orientations of elongating Pol II without and with RNase T1. The sliding distance is measured only from traces with a force rise above the baseline. The mean values (also as the red bars) and their SEMs are indicated. Sample sizes for peak force: CD: $N = 27$ (−RNase T1), $N = 19$ (+ RNase T1); HO: $N = 60$ (−RNase T1), $N = 21$ (+ RNase T1). Sample sizes for sliding distance: CD: $N = 23$ (−RNase T1), $N = 4$ (+RNase T1); HO: $N = 60$ (−RNase T1), $N = 21$ (+RNase T1). Source data are provided as a Source data file.

arms (Supplementary Fig. 2)[32]. This Y-template was unzipped with the unzipping fork approaching the Pol II either co-directionally or head-on (Fig. 1a). Using this method, we first examined a Pol II elongation complex paused at A20, containing 20-nt RNA complementary to the DNA template (with a total RNA length of 25 nt) ("Methods"), which should have a limited length of RNA outside Pol II for R-loop formation (Fig. 1b and Supplementary Fig. 3). We found that before the unzipping fork encounters Pol II, the unzipping force follows the naked DNA force baseline. When the fork encounters Pol II, the unzipping force deviates from the force baseline at the bound Pol II. In the co-directional orientation, a bound Pol II has a mean peak disruption force of 18 pN, 3 pN above the naked DNA unzipping force baseline (15 pN), with minimal sliding of Pol II along DNA under the influence of the unzipping fork (Fig. 1c). In contrast, in the head-on orientation, the unzipping force rises significantly above the baseline at the bound Pol II, with a mean peak disruption force of 35 pN, 20 pN above the baseline, indicating that Pol II significantly resists the DNA fork progression (Fig. 1c). Furthermore, the force rise persists for about 100 bp, consistent with Pol II sliding backwards along the DNA, resisting fork progression (Fig. 1c).

Notably, the potential for R-loop formation is minimal with the A20 complex. Despite this, Pol II imposes a stronger roadblock to a DNA fork when oriented head-on vs co-directionally. This polarity is evidenced by a significantly greater disruption force and longer sliding distance. The extended sliding behavior is unexpected given the limited RNA length, suggesting that Pol II can maintain its interaction with the DNA independently of an RNA–DNA hybrid.

To investigate whether an elongating Pol II is also a polar roadblock to the DNA fork, we unzipped DNA through Pol II in a buffer containing 1 mM NTPs after resuming transcription from A20 for about 40 s. Under this condition, Pol II transcribes approximately 600 nt on average before encountering the DNA fork (Fig. 1d). When the DNA fork encounters an elongating Pol II moving co-directionally, the mean disruption force is 27 pN (or 12 pN above the baseline) with minimal sliding of Pol II, suggesting that an elongating Pol II is more stable than the paused Pol II at A20 in the co-directional orientation (Fig. 1e). When the DNA fork encounters an elongating Pol II head-on, the mean disruption force is 36 pN, 21 pN above the baseline, again greater than that of the co-directional orientation, with high force persisting over about 300 bp (Fig. 1e). These data involving an elongating Pol II reinforce the findings from the paused Pol II and demonstrate that an elongating Pol II is also a more potent and persistent roadblock in the head-on orientation than in the co-directional orientation.

In addition, we found that an elongating Pol II is a stronger roadblock to the DNA fork than a paused Pol II at A20 in both the co-directional and head-on orientation (compare Fig. 1c, e). However, it remains unclear whether this increased resistance arises from the inherent stability of an elongating Pol II or from the presence of a long nascent RNA that may enhance Pol II anchoring to DNA. To differentiate between the two possibilities, we performed additional experiments with both a paused Pol II at A20 and an elongating Pol II in the presence of RNase T1, which digests the RNA outside the enzyme[33–35]. RNase T1 should effectively target the long RNA of an elongating Pol II and may also partially digest the 25-nt RNA of a paused Pol II at A20 (Supplementary Fig. 4). We found that for a paused Pol II at A20, the presence of RNase T1 slightly decreases the disruption force and sliding distance of both orientations. Interestingly, for an elongating Pol II, the presence of RNase T1 during transcription significantly decreases the disruption force for both orientations. The digestion of RNA also considerably reduces the sliding distance for the head-on orientation, making such a Pol II elongation complex behave more like a paused Pol II at A20. Thus, the presence of a long nascent RNA contributes to the enhanced resistance of Pol II to fork progression.

The progressing DNA fork used here somewhat resembles a progressing replication fork in that the DNA fork is mechanically unwound rather than being unwound by CMG. It is possible that such a simple system does not fully capture the intricacies of a replisome, which involves interactions among multiple proteins. Nonetheless, our findings might still provide some mechanistic insights into the polarity of the Pol II roadblock during transcription–replication conflicts. Because the Pol II roadblock polarity does not require a long RNA, the polarity should be inherent to an elongation complex. To remove the Pol II roadblock in either direction, the fork must progress through the elongation complex, disrupting interactions located both upstream and downstream of Pol II's active site. If the Pol II roadblock strength solely depends on the sum of these interactions, then the Pol II roadblock should have no polarity to a progressing fork. Instead, our findings suggest that the polarity depends on how the fork approaches an elongation complex. In the co-directional orientation, the DNA fork first encounters the upstream edge of the transcription bubble as indicated by the structure of the elongation complex[36–38]. The low disruption force from our work suggests that after the fork separates the two DNA strands in the bubble, Pol II's DNA clamp, downstream of the active site, becomes weaker, rendering the elongation complex unstable. In contrast, in the head-on orientation, the fork first encounters the front edge of Pol II downstream of its active site. The high disruption force from our work suggests that Pol II can tightly clamp onto the DNA downstream of its active site, strongly resisting destabilization by an advancing fork.

## RNA–DNA hybrid formation

The data in Fig. 1 show that a Pol II elongation complex is more stable against an advancing fork when associated with a long nascent RNA, raising the possibility that such stabilization is mediated by the formation of a long RNA–DNA hybrid. To investigate this possibility, we re-examined the co-directional unzipping traces (Fig. 1d). As shown in the more detailed view of an example trace (Fig. 2a), immediately after the unzipping fork encounters the bound Pol II and disrupts it, we detect an extension shift, where the force profile follows that of the naked DNA but at a shorter extension. In many traces, this extension shift occurs immediately upon the fork encountering Pol II (Supplementary Fig. 5). If this extension shortening is due to RNA–DNA hybrid formation, the shortening may increase with the RNA transcript length (Fig. 2b). If the entire RNA transcript forms a hybrid, the extension shortening is proportional to the RNA transcript length. Interestingly, the measured extension shortening agrees well with this prediction (Fig. 2c), indicating that the entire RNA transcript can readily form an RNA–DNA hybrid once the ssDNA becomes available near the bound Pol II.

We also re-examined head-on unzipping traces of an elongating Pol II, such as the one shown in Fig. 1d, in a more detailed view (Fig. 2d). In contrast to the co-directional encounter, the nascent RNA is located distal to the DNA fork. When the unzipping fork encounters the front of the bound Pol II (Fig. 2d), the force rises dramatically. After the unzipping force returns to the baseline and unzips past the transcription start site (TSS), we again detected an extension shortening, where the force profile followed that of the naked DNA at a shorter extension. This shortened extension may be indicative of the extent of the RNA–DNA hybrid on the lagging strand (Fig. 2e). We found that most of the traces show a hybrid formation consistent with an entire RNA transcript being converted to the hybrid (Fig. 2f). A minority of traces have a hybrid length shorter than expected, indicating that not all nucleotides of the RNA transcript are converted to a hybrid. This suggests that some regions of the RNA may be unavailable for hybridization in this orientation. Because the hybrid detection occurs after Pol II disruption, these data suggest that the RNA transcript can hybridize with the template strand even after the transcription elongation complex is disrupted.

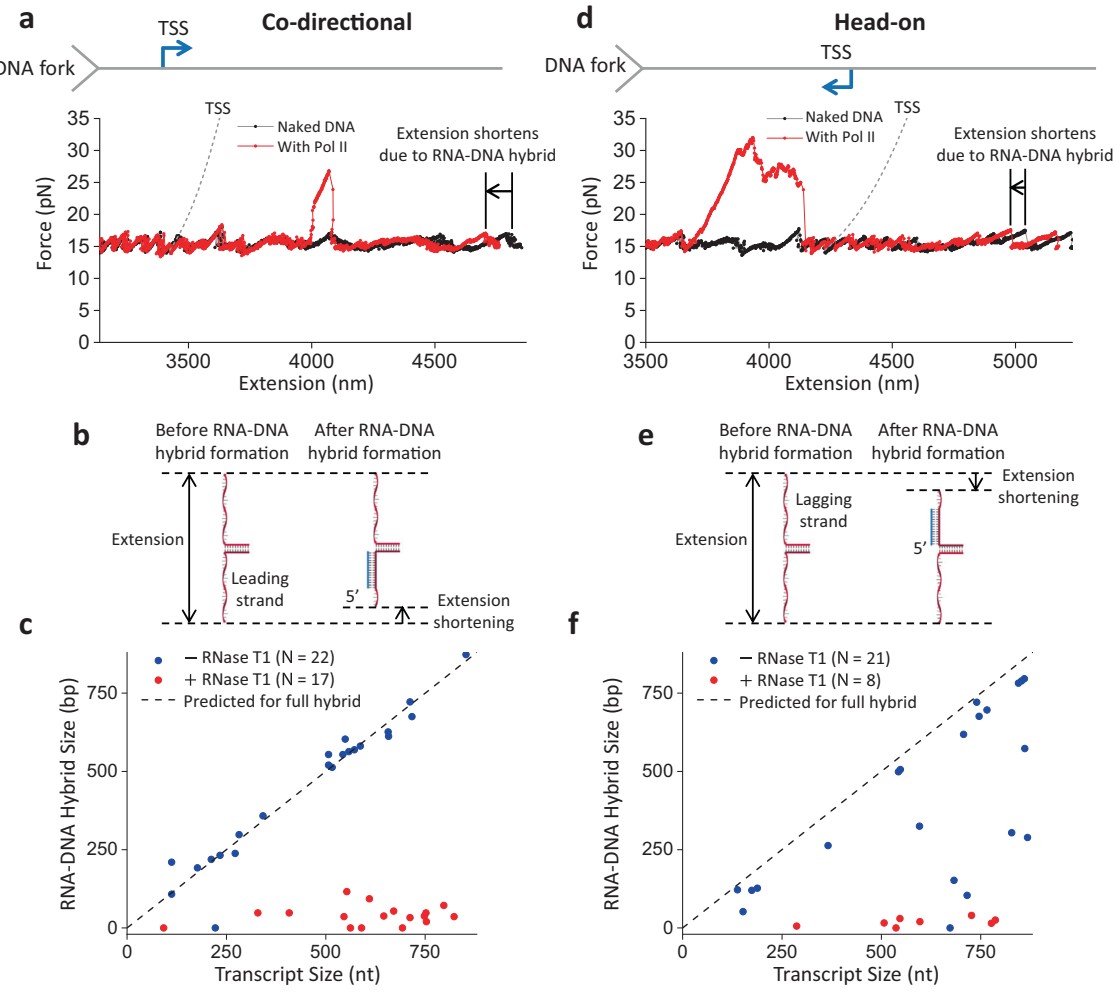

**Fig. 2 | RNA–DNA hybrid formation after DNA fork collision. a** Detailed view of a representative force-extension trace of the DNA fork unzipping through an elongating Pol II in the co-directional orientation, showing an extension shift, where the force profile follows that of the naked DNA but at a shorter extension. The dashed curve indicates the predicted force-extension curve for an unzipping fork encounter with a Pol II at the TSS. **b** Cartoon depiction of how an RNA–DNA hybrid leads to extension shortening in the co-directional orientation, with the hybrid forming on the leading strand. **c** RNA–DNA hybrid size, measured after the disruption of Pol II by the DNA fork, plotted against transcript size for the co-directional orientation. The dashed line indicates the predicted hybrid size if the entire transcript RNA is converted to a hybrid. (−) RNase T1, $N = 22$; (+) RNase T1,

$N = 17$. **d** Detailed view of a force-extension trace of the DNA fork unzipping through an elongating Pol II in the head-on orientation, showing an extension shift, where the force profile follows that of the naked DNA but at a shorter extension. The dashed curve indicates the predicted force-extension curve for an unzipping fork encounter with a Pol II at the TSS. **e** Cartoon depiction of how the RNA–DNA hybrid leads to extension shortening in the head-on orientation, with the hybrid forming on the lagging strand. **f** RNA–DNA hybrid size, measured after the disruption of Pol II by the DNA fork, plotted against transcript size. The dashed line indicates the predicted hybrid size if the entire transcript RNA is converted to a hybrid. (−) RNase T1, $N = 21$; (+) RNase T1, $N = 8$. Source data are provided as a Source data file.

Because RNA–DNA hybrid formation requires RNA, the presence of RNase T1 during transcription should minimize hybrid formation. Consistent with this prediction, when RNase T1 is present during transcription, DNA extension shortening after unzipping through Pol II is minimal in all traces for both the co-directional and head-on orientations (Fig. 2c, f and Supplementary Fig. 6). These observations further support that the observed DNA shortening is due to RNA–DNA hybrid formation, which can only occur in the presence of long RNA.

Importantly, we never detected any RNA–DNA hybrid before the DNA fork encountered Pol II, highlighting that the hybrid formation requires the presence of complementary ssDNA near Pol II. In vivo, when a replisome approaches a Pol II co-directionally, ssDNA immediately behind Pol II should rarely occur until the replisome encounters Pol II. Previous studies showed that Pol II generated (−) supercoiling behind could facilitate DNA melting and R-loop formation behind Pol II[39–44]. However, such an R-loop is less likely to form and sustain when a replisome trails behind Pol II because the (+) supercoiling generated by

the replisome can neutralize the (−) supercoiling from Pol II. An exception may arise if the replisome stalls at a lesion, leading to the decoupling of the replicative DNA polymerase and helicase[45,46]. In this case, the continued unwinding by the helicase may generate ssDNA[47] right behind the Pol II, enabling RNA–DNA hybrid formation. Thus, while R-loops can form in co-directional conflicts, they are expected to be less frequent. In contrast, when a replisome encounters Pol II head-on, the replisome progression may disrupt the transcription elongation complex but leave the RNA behind. Continued replication then exposes ssDNA on the lagging strand, providing a substrate for RNA–DNA hybridization.

### RNA–DNA hybrid in front of Pol II

Our finding that an RNA–DNA hybrid can form when complementary ssDNA is present in the Pol II vicinity raises the possibility that a hybrid may also form in front of Pol II during a head-on collision with a replisome in vivo. The replisome motor may be sufficiently strong to

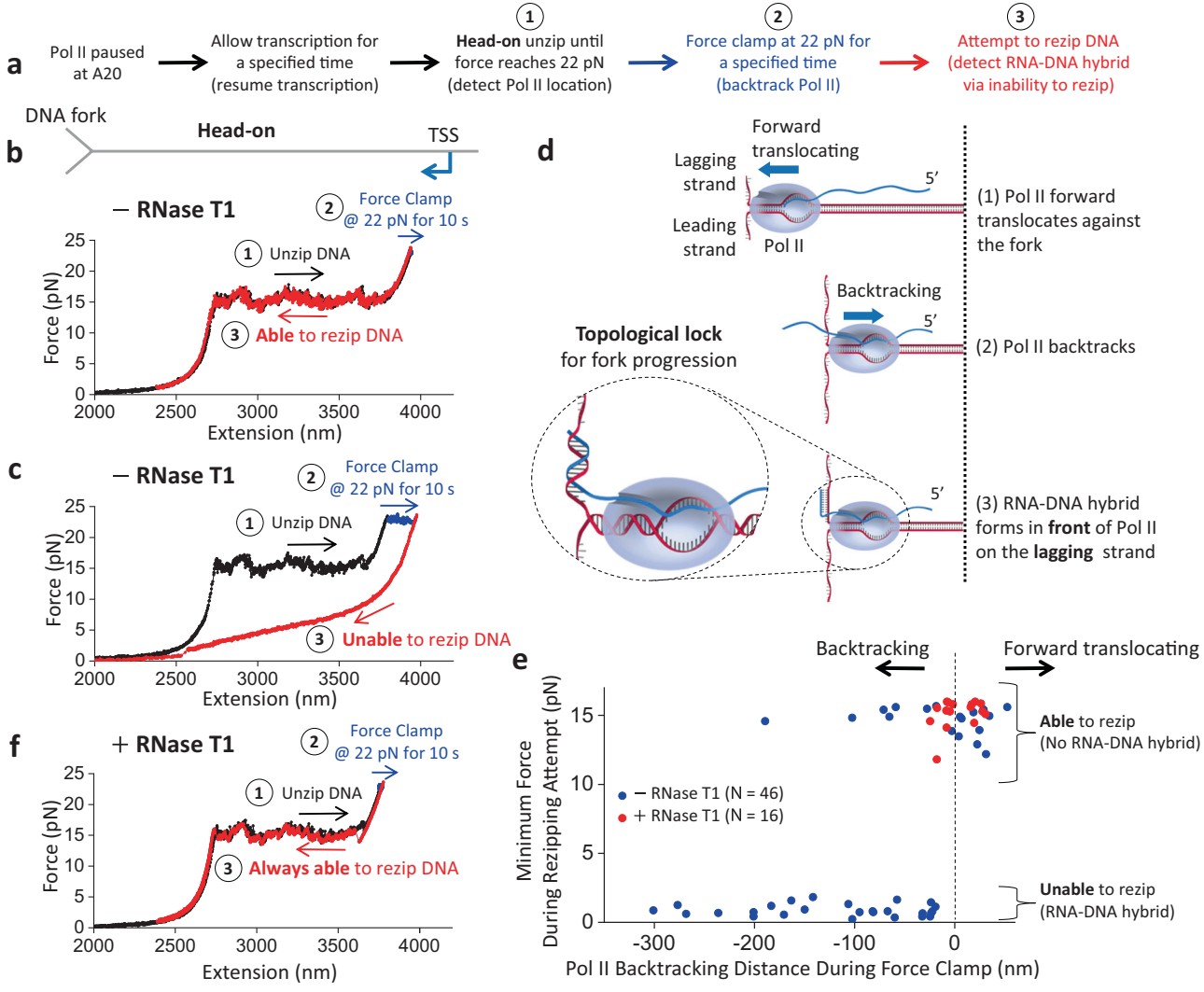

**Fig. 3 | RNA–DNA hybrid formation in front of Pol II. a** Outline of the experimental steps to backtrack Pol II and determine if RNA–DNA hybrid formation occurred. **b** Representative force-extension trace of the DNA fork interaction with an elongating Pol II in the head-on orientation, with minimal backtracking during the force clamp step. The DNA can be rezipped at step 3. **c** Representative force-extension trace of the DNA fork interaction with an elongating Pol II in the head-on orientation, with extensive backtracking during the force clamp step. The DNA cannot be rezipped at step 3. **d** Cartoon depicting the proposed fork and Pol II conformation after Pol II is backtracked upon head-on collision with the fork. The 3′ RNA extruded from a backtracked Pol II can hybridize with the lagging strand in front of Pol II. For ease of strand tracking, dsDNA or RNA–DNA hybrids are shown as two parallel strands, except for the inset where their helical structures are correctly

depicted. The inset illustrates how the RNA–DNA hybrid formation restricts Pol II rotation and DNA unwinding, topologically locking Pol II on the DNA. **e** The minimum force reached during the attempt to rezip step vs the amount of backtracking during the force clamp step, in the presence and absence of RNase T1. A low minimum force characterizes an inability to rezip. The black dashed line indicates the initial Pol II position before the force clamp. The backtracked distance in base pairs may be estimated using a conversion factor of 1.0–1.2 bp/nm. (−) RNase T1, $N = 46$; (+) RNase, T1 $N = 16$. **f** Representative force-extension trace of the DNA fork interaction with an elongating Pol II in the head-on orientation in the presence of RNase T1. The DNA can be rezipped at step 3. Source data are provided as a Source data file.

cause backtracking of Pol II, leading to 3′ RNA protrusion from Pol II's secondary channel[48–52]. This protruded RNA may hybridize with the lagging strand at the replication fork, where short stretches of ssDNA should be transiently present[53–55]. Thus, an RNA–DNA hybrid could form before Pol II disruption in a head-on transcription–replication collision.

To investigate this possibility, we mimicked the head-on transcription–replication collision using the DNA fork and an elongating Pol II but introduced an extra step to backtrack Pol II (Fig. 3a). Here, we allowed Pol II to transcribe for some distance before unzipping to Pol II. Once the DNA fork encountered Pol II, we held the unzipping force at 22 pN for 10 s to facilitate Pol II backtracking. The extent of the backtracking at this step varied from molecule to molecule, with some showing minimal backtracking (Fig. 3b) while others

showed extensive backtracking (Fig. 3c). Subsequently, we attempted to rezip the DNA by reducing the extension. We observed that when Pol II shows minimal backtracking, the DNA can be fully rezipped (Fig. 3b), but when Pol II shows extensive backtracking, the DNA typically cannot be fully rezipped, as evidenced by the decrease in force when the two DNA ends are brought closer (Fig. 3c).

Since DNA rezipping requires both ssDNA strands to be available for base pairing, the inability to rezip indicates an obstruction in the ssDNA strands. This could occur if an RNA–DNA hybrid forms on the lagging strand in front of a backtracked Pol II (Fig. 3d). If so, then an RNA–DNA hybrid may form more readily for more extensively backtracked Pol II since there is a greater opportunity for RNA–DNA hybridization. Consistent with this expectation, we found a strong correlation between the inability to rezip (characterized by a low

minimum force upon the rezipping attempt) and the backtracking distance (Fig. 3e), suggesting that the inability to rezip is an indicator of an RNA–DNA hybrid in front of Pol II. A control experiment conducted in the presence of 5 nM RPA demonstrates that an RNA–DNA hybrid can still form relatively efficiently, indicating that hybrid formation effectively competes with RPA binding to ssDNA (Supplementary Fig. 7).

To validate this interpretation further, we carried out the same experiments but in the presence of RNase T1, which can degrade RNA to limit Pol II backtracking (Fig. 3f). Without backtracking, the 3′ RNA will not protrude from Pol II's secondary channel to allow RNA–DNA hybrid formation in front of Pol II. Thus, the presence of RNase T1 should enable more efficient rezipping. Indeed, in the presence of RNase T1, we detected minimal backtracking, with all traces being fully rezipped (Fig. 3e).

The results in Fig. 3 provide substantial evidence for RNA–DNA hybrid formation in front of a backtracked Pol II at a DNA fork in a head-on orientation. The formation of such a hybrid further anchors Pol II to the DNA substrate. It is important to note that dsDNA and RNA–DNA hybrid assume helical structures, although our cartoons omit helicity to ease visual clarity of strand tracking. Due to this helicity, unwinding the DNA leads to a concurrent rotation of the parental DNA and, consequently, the bound Pol II. This rotation can readily occur until RNA–DNA hybrid formation, which restricts Pol II rotation and further DNA unwinding. As a result, RNA–DNA hybrid formation topologically locks Pol II onto the DNA (Fig. 3d inset), making it difficult for the unzipping fork to disrupt the bound Pol II. This mechanism likely underlies the increased Pol II resistance in the presence of a long nascent RNA, which enables the RNA–DNA hybrid formation in front of Pol II during head-on encounters, as shown in Fig. 1e.

### TFIIS and RNase H facilitate the release of topologically locked Pol II

During a head-on transcription–replication collision in vivo, a potential topological lock could be released if the 3′ RNA detaches from Pol II with the help of anti-backtracking factors, such as TFIIS, which facilitates Pol II cleavage of the 3′ RNA at the active site in the backtracked Pol II[51,52,56–60]. The cleaved 3′ RNA segment could then exit Pol II's secondary channel, severing the connection between Pol II and the RNA–DNA hybrid. Alternatively, the RNA–DNA hybrid formed in front of Pol II may be degraded by RNase H, which cleaves the RNA in an RNA–DNA hybrid and is known to mitigate R-loop formation and promote replication restart[61,62]. To investigate the potential roles of TFIIS and RNase H, we performed similar experiments to those in Fig. 3 in the presence of TFIIS or RNase H but with an additional step of attempting to unzip the DNA to detect the final location of Pol II (Fig. 4a). For these experiments, we focused on traces that backtrack significantly during the force clamp step and cannot be rezipped initially, consistent with RNA–DNA hybrid formation in front of Pol II.

We found that in the absence of TFIIS or RNase H, the DNA remains unable to be rezipped, and Pol II backtracks further from step 2 when examined during the final step to attempt unzipping (Fig. 4b top plot). This supports the possibility that the force on the DNA during the last two steps, even though small, could still promote Pol II backtracking. We observed this behavior even in the presence of TFIIS, suggesting that the TFIIS backtracking rescue time scale may be greater than our experimental time scale (~35 s). However, with TFIIS, we also detected traces that show a new behavior (Fig. 4b, middle plot): even though a trace cannot be rezipped initially, it subsequently becomes rezipped. In these traces, we found that Pol II regains its ability to translocate forward by up to a few hundred base pairs when we checked the Pol II position during the last step (Fig. 4c and Supplementary Fig. 8). This behavior is never observed without TFIIS. The emergence of this new behavior is consistent with the interpretation

that 3′ RNA cleavage of the RNA–DNA hybrid facilitates transcription reactivation and hybrid removal.

In the presence of RNase H, all traces that do not rezip initially become rezipped as the rezipping step proceeds, indicating that RNase H is highly efficient in cleaving the RNA in the hybrid so that the cleaved RNA fragments can be subsequently removed by a rezipping fork (Fig. 4b bottom panel; c). Furthermore, about 65% of these traces show that Pol II is reactivated and resumes transcription during the last step (Fig. 4c and Supplementary Fig. 8). In contrast to TFIIS-facilitated RNA cleavage that resets Pol II to an active elongation state, RNA cleavage by RNase H still leaves Pol II in a backtracked state, although the backtracked distance is greatly shortened, which may facilitate transcription resumption. Active transcription can only occur after Pol II escapes the backtracked state. Our data suggest this could occur spontaneously on our experimental time scale (~35 s).

Collectively, our data show that the presence of TFIIS or RNase H facilitates hybrid removal and transcription reactivation. These findings may have significant in vivo implications for a head-on Pol II collision with a replisome. Replisome progression may backtrack Pol II, leading to RNA–DNA hybrid formation on the lagging strand in front of Pol II. This hybrid could topologically lock Pol II on DNA, exacerbating the Pol II roadblock to the replisome. TFIIS or RNase H could facilitate cleavage of the 3′ RNA, which detaches the RNA–DNA hybrid from the bound Pol II and facilitates hybrid removal to alleviate the Pol II roadblock (Fig. 4e).

### RNA–DNA hybrid enables lagging-strand replication

If an RNA–DNA hybrid forms on the lagging strand, this hybrid can serve as a primer for lagging-strand replication. Demonstration of this possibility will provide an additional validation for the formation of RNA–DNA hybrid on the lagging strand. To explore this possibility, we extended our experimental approach used in Fig. 3 to enable lagging-strand replication (Fig. 5a). We first backtracked Pol II before attempting to rezip the DNA (Fig. 5b). If the force continued to drop during the rezipping attempt, the inability to rezip indicated the formation of the RNA–DNA hybrid. We then allowed the force to decrease to around 1 pN before transitioning to a buffer containing T7 DNA polymerase and dNTPs that allow lagging-strand replication ("Methods") (Fig. 5c, d). Under this force, one base pair of dsDNA has a longer extension than one nucleotide of ssDNA. This differential extension can be used to indicate lagging-strand replication. Figure 5c shows an example trace where the extension increases steadily with time, indicating steady lagging-strand replication. The extension increase stops at a position close to what would be expected, assuming all ssDNA on the lagging strand is converted to dsDNA (Fig. 5e). As a control, DNA extension changes minimally when the experiment is conducted without DNA polymerase (Fig. 5e).

Therefore, we demonstrate that an RNA–DNA hybrid formed on the lagging strand during a head-on collision between a DNA fork and Pol II can enable lagging-strand replication. Here, we used the T7 DNA polymerase for simplicity as a proof of principle. In a eukaryotic replisome, the formation of an RNA–DNA hybrid eliminates the need for priming, which could be used for subsequent lagging-strand synthesis by Pol α followed by Pol δ. This process does not require Pol II removal. Instead, it only requires Pol II to backtrack, which would be expected when Pol II collides with a replisome head-on. A backtracked Pol II then allows RNA–DNA hybrid formation in front of Pol II, and such a hybrid can then initiate lagging-strand replication even if Pol II remains bound.

## Discussion

In this work, we mimicked the replisome progression using mechanical unzipping of DNA. Using this approach, we investigated the consequences of an advancing DNA fork colliding with a transcribing Pol II either co-directionally or head-on. Our work provides a physical

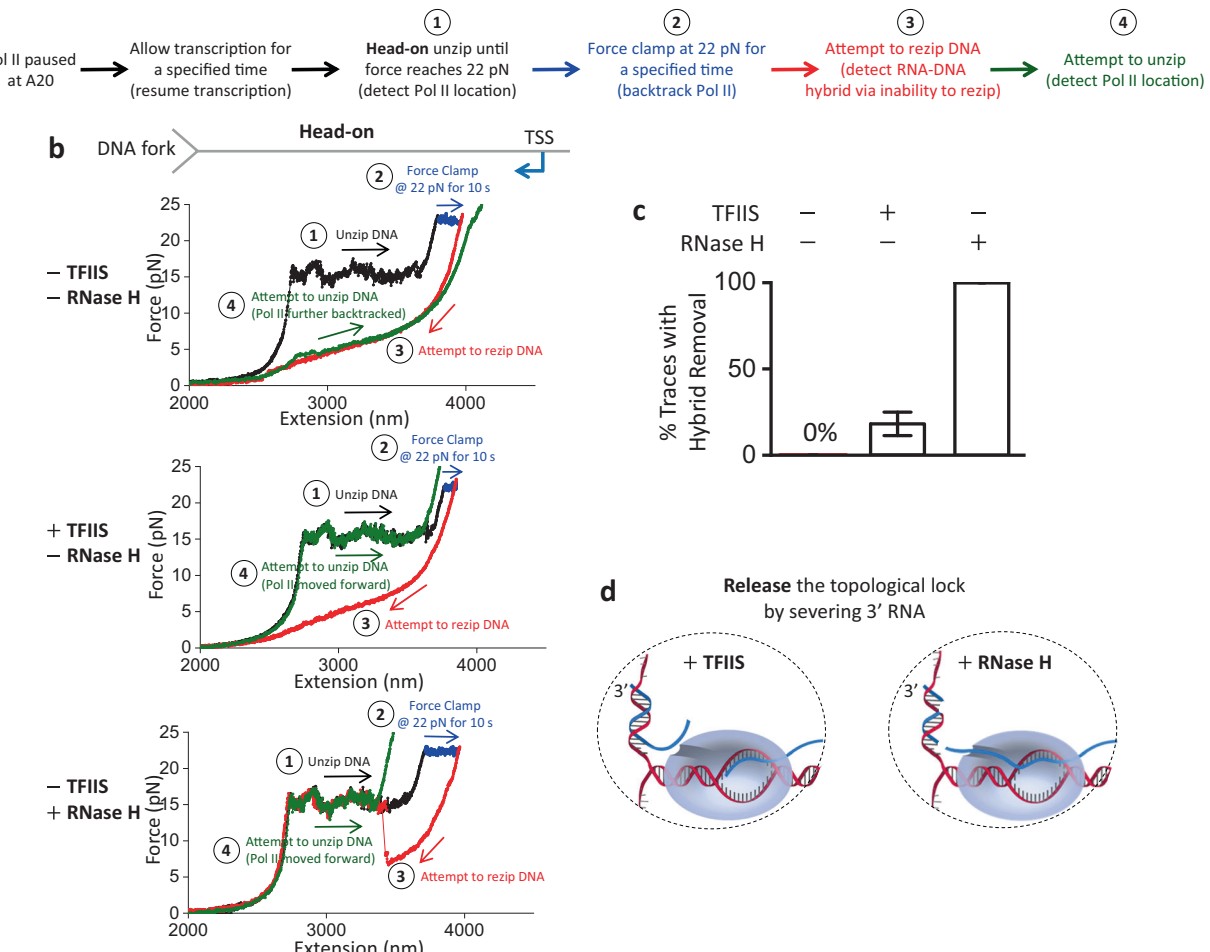

**Fig. 4 | TFIIS and RNase H facilitate RNA−DNA hybrid removal from backtracked Pol II. a** Outline of the experimental steps to backtrack Pol II, form RNA−DNA hybrid, and detect Pol II location after RNA−DNA hybridization. **b** Representative force-extension traces showing the interaction of the DNA fork with an elongating Pol II in the head-on orientation without TFIIS or RNase H (top plot), with TFIIS (middle plot), and with RNase H (bottom plot). For the trace without TFIIS or RNase H, after Pol II is backtracked in step 2, DNA cannot be rezipped in step 3. Pol II is further backtracked when checked during step 4. For the trace with TFIIS, after Pol II is backtracked in step 2, DNA cannot be rezipped in step 3. However, DNA becomes rezipped (indicating hybrid removal), and Pol II has forward translocated when checked during step 4. For the trace with RNase H, after Pol II is backtracked in step 2, DNA cannot be rezipped initially but becomes rezipped subsequently (indicating hybrid removal) during step 3. Pol II has also forward translocated when checked during step 4. **c** Percentage of traces that showed hybrid removal. The error bars represent SEMs of the counting errors (−TFIIS and −RNase H, with $N = 26$; +TFIIS, $N = 33$; +RNase H, $N = 20$). **d** Cartoons illustrating how TFIIS and RNase H can facilitate the release of the topologically locked Pol II by severing the 3′ RNA. Source data are provided as a Source data file.

explanation for the polarity of the Pol II roadblock, demonstrates that an RNA−DNA hybrid can form not only behind Pol II but also in front of Pol II, and raises the possibility that the RNA−DNA hybrid formed in front of Pol II has the potential to initiate lagging-strand replication.

We show that Pol II roadblock polarity to the DNA fork is inherent to the Pol II elongation complex (Fig. 1): Pol II can greatly resist fork progression by sliding along the DNA in the head-on orientation, even when the transcript size is short. This intrinsic polarity could explain why transcription−replication conflicts are more likely to induce replisome stalling during a head-on transcription−replication conflict in the cell. Our unzipping mapper reveals that in the co-directional orientation, the Pol II elongation complex is unstable and can be readily disrupted; however, in the head-on orientation, it is highly stable and strongly resists disruption. A similar polarity has also been shown for *E. coli* RNAP elongation complex[27,29,32] (Supplementary Fig. 9), suggesting that this polarity may be inherent to all transcription elongation complexes. This inherent polarity may help explain the polarity of the RNAP roadblock to replication for both eukaryotes and prokaryotes.

Although the Pol II roadblock has an inherent polarity, we found that the presence of a long RNA transcript strengthens Pol II's interactions with DNA in both orientations while retaining the Pol II roadblock polarity (Fig. 1). We show that an elongating Pol II with a long RNA transcript is also a more potent and persistent roadblock in the head-on orientation than in the co-directional orientation. We observed that a long RNA transcript allows RNA−DNA hybrid formation if there is available ssDNA near the Pol II (Fig. 2). Significantly, we made a surprising discovery that during a Pol II head-on collision with the DNA fork, an RNA−DNA hybrid can form on the lagging strand in front of Pol II after Pol II backtracks (Fig. 3). The enhanced Pol II interaction with DNA is likely a result of an RNA−DNA hybrid anchoring Pol II to the DNA via a topological lock (Fig. 3d). Consistent with this interpretation, the presence of TFIIS, which facilities Pol II cleavage of the 3′ RNA, allows RNA−DNA hybrid removal, suggesting that the connection between the bound Pol II and the RNA−DNA hybrid is severed. Interestingly, while the prevalent view places an RNA−DNA hybrid behind Pol II during a transcription−replication conflict[25], recent studies show that the presence of TFIIS is critical to maintaining genome stability and allude to the possibility of the formation of an RNA−DNA hybrid in front of Pol II[63,64]. Our data now provide additional evidence for these emerging views. Further supporting this interpretation, we found that RNase H is highly efficient at facilitating the

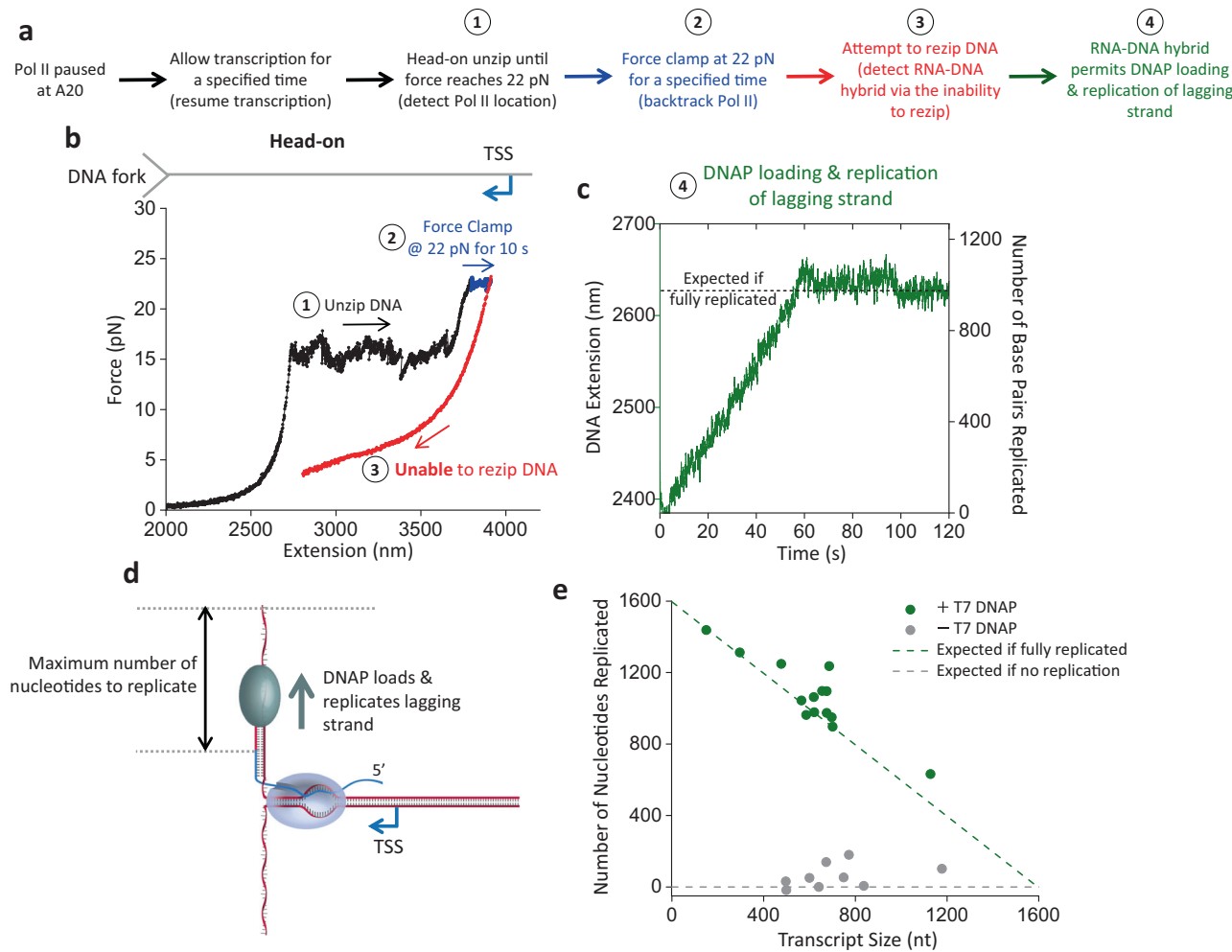

**Fig. 5 | RNA–DNA hybrid formation enables lagging-strand replication.**
**a** Experimental outline for steps to locate and backtrack Pol II, detect RNA–DNA hybrid formation, and observe T7 DNAP replication on the lagging strand.
**b** Representative force-extension trace of the DNA fork interaction with an elongating Pol II in the head-on orientation, with extensive backtracking. **c** Monitoring the real-time DNA replication during step 4 of the trace shown in (**b**). During this step, the force clamp is held at around 1 pN. Under this low force, DNA extension increases as T7 DNAP converts the ssDNA of the lagging strand into dsDNA. The right vertical axis shows the conversion from extension into the number of base pairs replicated. The black dashed line represents the position if T7 DNAP fully replicates the available lagging strand. **d** Cartoon of T7 DNAP replication of the lagging strand using the RNA–DNA hybrid as a primer. **e** Replication distance vs transcript size. The green dashed line indicates the expected number of nucleotides replicated if the lagging strand is fully replicated. (+) T7 DNAP, $N = 14$; (−) T7 DNAP, $N = 9$. Source data are provided as a Source data file.

removal of such a hybrid in front of Pol II, also leading to the release of such a topological lock. This finding provides an explanation for the crucial role of RNase H in mitigating the RNAP obstacle during a head-on transcription–replication conflict.

In vivo, RNA–DNA hybrids have indeed been found on the lagging strand behind a eukaryotic replication fork in head-on collisions[23,65,66]. Our work may provide a mechanistic explanation for this observation. Ultimately, the stalled Pol II must be moved off the DNA to allow replication progression[16]. It is possible that the replisome, especially with the help of other helicases, such as Pif1 and Rrm3[67,68], is sufficiently powerful to evict the stalled Pol II. Factors directly interacting with Pol II could also assist Pol II eviction during such collisions, but they are yet to be identified. Although Mfd in prokaryotes can evict an RNAP stalled at a DNA fork[16,27,69–71], its eukaryotic counterparts, CSB (Human) or Rad26 (*S. cerevisiae*), do not exhibit this capability[72,73].

We hypothesize that an RNA–DNA hybrid can form in front of Pol II only after Pol II physically encounters the replication fork during a head-on transcription–replication conflict. However, when Pol II approaches a replisome head-on, (+) torsion accumulates between Pol II and the replisome well before Pol II physically encounters the replisome, because torsion in the DNA can act over a distance[21]. This

(+) torsion accumulation could induce replication fork stalling and disassembly[74], as well as Poll II backtracking and stalling[21,39,75]. Topoisomerases can resolve torsional stress during these conflicts, but their impact on the RNA–DNA hybrid formation during these conflicts is a double-edged sword. Before Pol II encounters a replisome, supercoiling relaxation allows Pol II progression and limits its backtracking, potentially reducing the RNA–DNA hybrid formation at the fork once the collision occurs. However, supercoiling relaxation increases the opportunity for collision by allowing the replisome and Pol II to come into direct contact, which then permits RNA–DNA hybrid formation at the fork. Previous studies of the head-on transcription–replication conflict found that supercoiling resolution by gyrase in prokaryotes drives RNA–DNA hybrid formation[75], whereas supercoiling resolution by topoisomerase I in eukaryotes prevents RNA–DNA hybrid formation[5,43,76,77].

Our work uses a DNA fork to mimic the replisome and may not capture the full complexity of what might occur during transcription–replication conflicts. However, the simplicity of our model system permits mechanistic studies and reveals important physical parameters for the formation of the RNA–DNA hybrid. We now have a significantly clearer understanding of the nature of the

RNA–DNA hybrid. We anticipate that what we have learned from this work may facilitate data interpretation of more complex in vivo systems.

## Methods

### Protein expression and purification

Yeast RNA Polymerase II (Pol II) containing a 6xHis-tag on both Rpb1 and Rpb3 subunits was expressed and purified from *S. cerevisiae*[78]. In brief, cell pellets were lysed and applied to a HisTrap HP column (Cytiva 17524801), a heparin column (Cytiva 17040703), and a Mono Q column 10/100 (Cytiva 17516701). The eluted Pol II was assessed via SDS-PAGE 4–12% Bis-Tris SDS gels in 1X MES SDS buffer, then concentrated and flash-frozen for storage.

*S. cerevisiae* recombinant TFIIS was expressed in *BL21 (DE3) pET15b PPR1* (69450-M; Novagen-sigma)[79] Cell pellets were harvested by centrifugation and then sonicated using a microprobe tip and Sonicator-Ultrasonic Processor VHX 750 watt (model GEX 750; PG Scientific). Each chromatography step was assessed via SDS-PAGE with 12% Bis-Tris SDS gels in 1× MES SDS buffer. The centrifuged lysate was applied to a HisTrap FF Crude column (Cytiva 17528601) and then applied to a Mono S column 10/100 GL (Cytiva 17516901). The eluted protein was diluted and applied to a HisTrap HP (Cytiva 29051021) column and a Mono S column 5/50 GL (Cytiva 17516801) to produce a 6xHis-tagged protein. The protein was assessed via SDS-PAGE, concentrated, and flash-frozen for storage.

RPA1, RPA2, and RPA3 were co-expressed in *E. coli* BL21 (DE3) from plasmid pJM1263[80]. In brief, cell pellets were lysed by sonication, and clarified lysates were applied to a HiTrap Blue HP column, followed by ssDNA cellulose (Worthington), and MonoQ chromatography. The eluted protein was dialyzed and flash-frozen for storage.

Wild-type T7 DNA polymerase was purchased from New England Biolabs (NEB, M0274S). *E. coli* thioredoxin (trx) was purchased from Sigma-Aldrich (Sigma, T0910). RNase T1 was purchased from Thermo Fisher Scientific (Thermo Fisher, EN0542). RNase H was purchased from New England Biolabs (NEB, M0297L).

### DNA substrates

We utilized a Y-shaped DNA substrate mimicking a DNA replication fork[26–30,32,81,82] (Supplementary Fig. 2) that enabled DNA unzipping of a parental strand with Pol II assembled in the elongation complex. This Y-shaped DNA structure consists of Y-arms (fork-like structures with two daughter strands) and a parental DNA strand with an assembled Pol II elongation complex.

The Y-arms consist of both leading and lagging daughter strands. Two 4.15 kb daughter strands of identical sequence were amplified via PCR from pBR322 (NEB N3033S). The leading strand was amplified with a forward primer containing a 5′ digoxigenin (dig) label, while the lagging strand was amplified with a forward primer containing a 5′ biotin (bio) label (Supplementary Table 1). Each resulting arm was digested with BsmBi-V2 (NEB R0739S) and ligated with T4 DNA Ligase (NEB M0202) to the respective leading (upper leading strand and lower leading strand) or lagging (upper lagging strand and lower lagging strand) strand annealed adapter oligo (Supplementary Table 1). The "lower" leading and lagging adapter oligos contain a 30 nt sequence of complementary ssDNA, allowing the two pieces to be annealed to one another and create the Y-arms.

Figures 1–4 used a Y-arm structure with no additional modifications (upper leading strand), while Fig. 5 had a modified leading strand adapter oligo (Upper Leading Strand_invdT) with an inverted dT on the 3′ end to prevent T7 DNAP from loading and replicating the leading strand.

The complete Y-shaped DNA substrate was created by ligation of the Y-arms to a parental DNA strand via a unique 3′ overhang. Two parental strands were generated for this study: co-directional (CD) or head-on (HO). Both strands consisted of three key pieces: the

transcription elongation complex, a downstream segment, and an upstream segment.

Both templates used an identical elongation complex DNA segment, consisting of four annealed complementary ssDNA oligos assembled with RNA Pol II[31,83,84]. Annealing resulted in a short 130 bp DNA construct with two unique overhangs to allow for ligation to the downstream and upstream templates. A downstream non-template strand (DS NTS), an upstream NTS (US NTS), and a template strand (TS) (Supplementary Table 1) were first annealed to create a 'gapped' DNA construct, with a 76 nt gap exposing the TS as ssDNA between the two NTS pieces. A 14 nt RNA (RNA 14), with 9 nt complementary to the TS, was annealed to the gapped region, forming an RNA–DNA hybrid scaffold. Pol II was added at a 3:1 molar ratio relative to the hybrid and incubated at room temperature before the addition of a complementary 76 nt NTS ssDNA oligo (ARP77). The nicks at the two ends of the gap were ligated by T4 DNA Ligase (M0202) in the same reaction listed below that ligates the downstream, upstream, and Y-arm structure together.

For the CD parental strand, the upstream segment was amplified from pBR322 (NEB N3033S) and enzymatically digested with both PpuMI (NEB R0506) and DraIII-HF (NEB R3510S) to generate a 0.65 kb strand. The downstream segment was amplified from pRL574 and digested with BstXI (NEB R0113), generating a 1.5 kb strand. The pRL574 plasmid, containing the partial beta subunit gene, was used to allow for a readable transcription sequence, but the T7 A1 promoter and the T7 TSS were not in the amplified region to avoid sequence overlap with the Pol II elongation complex sequence. The upstream segment was ligated to the Y-arms and the upstream end of the elongation complex, and the downstream segment was ligated to the downstream end of the elongation complex.

For the HO parental strand, which was assembled in the opposite direction, the upstream segment was amplified from pBR322 (NEB) but only digested with PpuMI (NEB R0506), generating a 1.4 kb strand. The downstream template was amplified from the same region of pRL574 as the CD template but digested with both BstXI (NEB R0113) and AlwNI (NEB R0514S), generating a 1.5 kb strand. The upstream segment was ligated to the upstream end of the elongation complex, and the downstream segment was ligated to the Y-arms and the downstream side of the elongation complex.

For each CD or HO template, the Y-arm structure, appropriate corresponding downstream and upstream segments, and the elongation complex were ligated together using T4 DNA Ligase (NEB M0202) in the same reaction.

### Experimental conditions

All experiments were conducted in a multi-channel laminar flow cell at room temperature (23 °C). Figures 1–4 used a four-channel flow cell, while Fig. 5 used a five-channel flow cell. In the four-channel flow cell, Channel 1 contained anti-digoxigenin-coated polystyrene beads (1 μm, Polysciences, 08226-15) pre-incubated with a DNA Y-structure template with Pol II elongation complex pre-assembled on the DNA. The final DNA concentration used in the experiments was 2.5 pM. Beads were suspended in a 1× Transcription Buffer (TB150) (25 mM Tris-HCl pH 8.0, 150 mM KCl, 10 μM ZnSO4, 1 mM DTT, 2 mM TCEP, 3% Glycerol) with 1 mM MgCl2. Channel 2 contained A20 Buffer (1× TB150 with 4 mM MgCl2, 1 mM ATP, 1 mM GTP, and 1 mM CTP) for Pol II elongation to an Adenine (A) 20 nt away from the TSS. Channel 3 contained NTP Buffer (1× TB150 with 5 mM MgCl2, and 1 mM of each NTP) for full transcription. Both channels 2 and 3 contained an oxygen scavenger system (10 nM protocatechuate-3,4-dioxygenase and 2.5 mM protocatechuic acid) to minimize photodamage to the DNA and increase tether lifetime. Channel 4 contained streptavidin coated polystyrene beads (1 um, Polysciences, 08226-15) in 1× PBS (137 mM NaCl, 2.7 mM KCl, 8 mM Na2 HPO4, and 2 mM KH2PO4).

Variations in Channels 2 and 3 were present for different experimental conditions. In Figs. 1–3, (−) RNase T1 conditions had 0.25 u/μL of SUPERase·In RNase Inhibitor (Thermo Fisher AM2696). For data taken in a (+) RNase T1 environment, channel 3 contained 10 u/μL of RNase T1 (Thermo Fisher, EN0541). In Fig. 4, for data taken in the (+) TFIIS environment, and Supplemental Fig. 7, for data taken in the (+) RPA environment, Channel 3 was preincubated with 1 mg/mL acetylated BSA in 1× PBS, then exchanged to NTP buffer with 1 μM of TFIIS or 5 nM RPA, respectively. In Fig. 4, for data taken in the (+) RNase H environment, Channel 3 contained 0.2 u/μL of RNase H (NEB, M0297L).

In the five-channel flow cell, Channels 1–3 were identical to those in the four-channel flow cell, with Channels 2 and 3 containing both the oxygen scavenger system and 0.25 u/μL of SUPERase·In RNase Inhibitor. Channel 4 contained 1× replication buffer (RB) (50 mM Tris-HCl (pH 7.5), 40 mM NaCl, 1.5 mM EDTA, 8 mM MgCl$_2$, 1 mM DTT, 2 mM TCEP, and 3% glycerol) with 1 mM of each dNTP and 0.5 mg/mL β-casein, along with the oxygen scavenger system. In the presence of T7 DNAP, 20 nM wt T7 DNA polymerase and 150 nM thioredoxin were included. Channel 5 contained streptavidin-coated polystyrene beads (1 μm) in 1× PBS.

## Single-molecule optical tweezers assay

Data was collected using a home-built dual optical trap in combination with a microfluidic multi-channel laminar flow cell[27,32]. Each channel of the flow cell was fed with a 1 mL glass syringe (Hamilton, 81320), driven by a syringe pump with a constant flow of 1.5 μL/min. The four-channel flow cell had a flow rate of 170 μm/s, and the five-channel flow cell had a flow rate of 210 μm/s at the trapping position.

The DNA template was tethered between two optically trapped beads via its labeled daughter strands. A strep-coated polystyrene bead was trapped in a steered trap and tethered via the 5′ biotin label on the lagging daughter strand, while an anti-dig-coated bead, preincubated with DNA, was tethered via the 5′ dig label on the leading strand. Tether formation occurred in channel 2, allowing Pol II to escape to an A20 pause site. The tether was then transferred to channel 3 to begin data collection.

## Single-molecule experimental procedures

All experiments began by mechanically unzipping the parental DNA at 100 nm/s to mimic the replication fork. Traces shown in Figs. 1 and 2 were unzipped until the end of the template was reached, thereby unzipping through and probing the fork interaction with the Pol II. Figures 3–5 unzipped the duplex DNA until the Pol II molecule was encountered. This was determined by the force reaching 7 pN above the naked DNA baseline. A constant force (7 pN above the DNA baseline) was applied for 10 s to induce backtracking. The ability to and the extent of backtracking varied between molecules. Experiments described in Figs. 3–5 tested for RNA–DNA hybrid formation by the inability to reanneal the duplex DNA in front of the Pol II after backtracking. This was tested by mechanically rezipping the DNA at a velocity of −100 nm/s. In Fig. 4 a subsequent DNA unzipping step (100 nm/s) was used to determine RNA–DNA hybrid removal and the final position of Pol II on the DNA template. In Fig. 5, after an RNA–DNA hybrid was identified, tethers were moved into Channel 4 and held at a constant force of around 1 pN for 2 min to detect T7 DNAP polymerase replication of the lagging strand.

## Data acquisition and data conversion

Data was acquired at 10 kHz and converted into force and extension as previously described[85]. Raw force and extension data were used to obtain the Pol II location and RNA–DNA hybrid size (bp) via the elasticity parameters for dsDNA[86], RNA–DNA hybrid[87], and ssDNA[88–90], as shown previously. Due to bead size variations, the trap stiffness of each bead pair was calibrated using the theoretically predicted force-extension curve and the measured unzipping force preceding the DNA fork's encounter with a bound Pol II.

## Unzipping alignment

To improve the precision and accuracy of the unzipping data, the extension of each bead pair was shifted and stretched to align with the theoretically predicted force-extension curve, using regions of the unzipping data that preceded the encounter of the DNA fork with Pol II[20].

## Single-molecule data analysis

The transcript size of the nascent RNA was determined by identifying the position where the DNA fork first encountered Pol II, relative to the known TSS. Pol II's location was inferred by force deviations from the theoretical naked DNA force-extension curve. Under co-directional elongation conditions with (+) RNase T1, where force-extension deviations were minimal, an additional rezipping step was used to locate the Pol II position. This was identified by a failure of the DNA duplex to reanneal, indicating a physical block by Pol II. The Pol II encounter extension was converted to base pairs unzipped using the freely-jointed chain (FJC) model, allowing the transcript size to be measured. The maximum disruption force was determined by measuring the maximum force reached while the fork interacted with Pol II, before complete disruption.

Pol II sliding distance was quantified by measuring the total distance required to disrupt its interaction with dsDNA. For traces with force peaks exceeding 18 pN, the initial Pol II encounter was refined using the fork dwell time histograms to minimize contributions from DNA stretching prior to sliding. In cases where RNA–DNA hybrid formation caused large extension snapbacks during the initial force rise, the sliding start was defined immediately after the hybrid formed. The end of the sliding was identified as the return to the naked DNA force baseline. Traces lacking a clear force peak or dip, and showing only extension shortening, were excluded from sliding distance analysis.

To assess whether RNase T1 significantly alters the roadblock properties of the Pol II elongation complex shown in Fig. 1c, e and Supplementary Fig. 4, unpaired two-sample $t$-tests (ttest2 in MATLAB, with 95% confidence interval for the mean difference) were performed comparing (+) and (−) RNase T1 conditions. These tests were applied to both the maximum disruption force and the sliding distance. For the force measurements, significant differences were observed in HO A20 ($p = 0.03$), CD Elongating ($p = 0.03$), and HO Elongating ($p = 0.01$), while CD A20 and HO RNA 14 showed no significant change ($p = 0.40$ and $p = 0.56$, respectively). For the sliding distance measurements, significant differences were found in HO A20 ($p = 0.02$) and HO Elongating ($p = 0.02$), whereas CD A20, CD elongating, and HO RNA14 did not show significant changes ($p > 0.22$). These results suggest that RNase T1 has an RNA-size-dependent effect on roadblock properties.

RNA–DNA hybrid formation size was measured by analyzing the shift in the number of unzipped base pairs from the predicted force-unzipped base pairs curve for the given DNA sequence. After the first encounter with Pol II, a shift was applied to align the data with the DNA baseline. For co-directional traces, this shift was aligned directly after Pol II disruption, while the head-on was shifted after the TSS site to ensure only the upstream DNA segment was being aligned. This shift was then used to calculate the RNA–DNA hybrid size using the FJC model for the ssDNA component and the WLC model for the RNA–DNA hybrid component.

The Pol II backtracking distance was determined by the difference between the initial and final Pol II locations during the force-clamp step. The minimum force during the rezipping attempt was identified as the lowest force value reached before the force-extension curve returned to the theoretical DNA baseline of naked DNA. The starting force upon unzipping attempt in step 4 of Supplementary Fig. 7 was calculated as the measured force at an extension of 2750 nm.

T7 DNAP replication was measured by the extension change that occurred during the 1 pN constant force clamp step. The overall extension was adjusted by subtracting off the extension contributions from the leading strand, Pol II backtrack distance, and RNA–DNA hybrid components, leaving the extension of the lagging strand available to be replicated. Due to secondary structures in the ssDNA[15], the FJC model cannot be used for converting ssDNA into nucleotides at a low force. Instead, a conversion factor (28 nt/nm) was measured in a (−) T7 DNAP environment around a force of 1 pN. The lagging strand extension was then converted into the number of nucleotides replicated using the WLC for the dsDNA and the measured conversion factor for the ssDNA.

## Quantification and statistical analysis

All data were obtained from at least eight independent replicates. Statistical details of individual experiments, including the number of traces, mean, and SEM values, can be found in the manuscript text, figures, and figure legends.

## Reporting summary

Further information on research design is available in the Nature Portfolio Reporting Summary linked to this article.

## Data availability

All data are provided in the main text and its supplementary files. Source data are provided with this paper.

## Code availability

Data analysis routines used to process and generate plots are available in the GitHub repository at https://github.com/WangLabCornell/Kay_et_al.[91].

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

## Acknowledgements

We thank members of the Michelle Wang Laboratory for helpful discussion and comments. We also thank Drs. J. Oh and P. Hou for purifying the 10-subunit Pol II used for initial troubleshooting of the elongation complex assembly. This work was supported by the National Institutes of Health grants R01GM136894 (to M.D.W.), T32GM008267 (to M.D.W.), R01GM102362 (to D.W.), and R35GM152094 (to D.R.). This work was also supported by the Intramural Research Program of the National Institutes of Health, National Cancer Institute, Center for Cancer Research (to M.K.). M.D.W. is a Howard Hughes Medical Institute Investigator.

## Author contributions

T.M.K., J.T.I., T.T.L., and M.D.W. designed single-molecule assays. T.M.K., J.Q., and P.M.H. designed DNA substrates. T.M.K. prepared DNA substrates. L.L. and M.K. purified and characterized the 12-subunit Pol II and TFIIS. S.B. and D.R. purified the RPA. T.M.K. performed single-molecule experiments. J.Q. and P.M.H. provided technical advice. T.M.K. and J.T.I. analyzed data. T.T.L. performed some preliminary experiments using *E. coli* RNA polymerase. D.W. provided the 10-subunit Pol II used only for initial troubleshooting of the elongation complex assembly. M.D.W. wrote the initial draft. All authors contributed to the manuscript revision. M.D.W. supervised this project.

## Competing interests

The authors declare no competing interests.
