## [Transparent Peer Review file · Nature Communications]

RNA Polymerase II is a Polar Roadblock to a Progressing DNA Fork

Corresponding Author: Professor Michelle Wang

Version 0:

Reviewer comments:

Reviewer #1

(Remarks to the Author)

Review of NatComm RNAP against unzipping block

This work from a leader in the field of single-molecule based investigations of protein-nucleic acid systems describes interesting experiments aimed at understanding what happens when an RNA polymerase encounters a strand-separated fork. They probe different directionalities of the encounter with respect to the direction of transcription and show that RNA-DNA hybrids may form between the nascent transcript and one or the other of the ssDNAs in the fork. They also show that these hybrids increase the block to fork progression; especially one formed in front of a backtracked RNAP. As expected, TFIIIS and RNase T1 release this extra blockage by cleaving the RNA. Lastly, they show that this RNA-DNA hybrid may serve as a primer for T7 DNA polymerase.

The experiments are very well done and well controlled. The paper is also clearly written, and it is straightforward to follow the logic presented. The results clearly augment our understanding of the molecular outcomes of the studied system.

My only concerns are based on what I feel is a bit of stretching as to the significance and interpretation of the data which I detail below:

(1) As the authors themselves introduce, we already know about the different outcomes of head-on and trailing collisions. In fact, 10 references are provided which describe this. Thus, what is the actual gap in knowledge that is being pursued in this work?

The authors claim that they are after a “fundamental mechanism” underlying this effect of directionality. However, when rationalizing why a head-on is different than a trailing collision, they re-state observations made from previous structural studies dealing with the strength of upstream and downstream interactions between the polymerase and the DNA. This mechanism could have been done without any of the single-molecule data. At the end of page 4, they write:

“Based on the Pol II EC crystal structure and supported by these data, Pol II interacts weakly with dsDNA behind its active site but tightly with dsDNA in front of its active site. These structural features provide a possible explanation for the Pol II polarity detected by the DNA fork and might also hold for a replisome.”

In my opinion, the initial single-molecule data simply confirms previous observations which takes away a bit of its significance.

In contrast, the observation that seems a bona fide new piece of information revealed by this specific approach is the topological lock that may form with the head-on collision, backtracking RNAP.

(2) Is a mechanically unwound fork a good fill in for a replisome? It seems taken for granted that the experimental setup reasonably mimicks a replisome. However, there are almost certainly caveats to this. In the context of a replisome, with the leading replicative helicase out in front, the accessible ssDNA would be significantly farther away from the polymerase itself and would not interact directly. The actual interaction would probably be mediated via protein:protein interactions between the helicase and the polymerase. To this point, the authors themselves comment in the text (again at the end of page 4) that:

“Since the progressing DNA fork used here somewhat resembles a progressing replication fork, our observations might provide a mechanistic explanation for previous findings that head-on transcription-replication conflicts are more detrimental than co-directional ones.”

Please provide clarification as to what “somewhat resembles” means. It sounds like a pretty strong caveat that merits discussion.

(3) As an add on to the previous point, it seems unlikely that in a replisome the ssDNA strands are splayed out at 180 degrees from one another. This geometry artificially brings the ssDNA closer to the polymerase. I wonder what the effect of holding the strands a different (more acute) angle would have with the prevalence of RNA-DNA hybrid formation. In principle, this could be done with the tweezers, but I understand it may represent a considerably amount of work.

(4) As a second add on to the degree to which the splayed fork resembles a replisome, an actively replicating system is known to have the ssDNA coated with single-stranded DNA binding proteins. It would seem straightforward that SSBs would inhibit the formation of the RNA-DNA hybrids revealed here.

Relatedly, even in the absence of SSBs, without the ssDNA held under tension, it would have the propensity to form secondary structures which may also prevent (or decrease) the formation of RNA-DNA hybrids.

(5) The data convincingly show that an RNA primer enables polymerization. Very cool experiment, but as the replisome already needs to exist in order to collide with the polymerase, its not clear that this behavior would ever contribute to replication. This experiment seems more confidently interpreted as another control that in fact, the RNA-DNA hybrid is forming where the authors say it is rather than a regulatory or biologically important point.

The authors do say that this is a proof of principle only due to the use of T7 instead of pol delta, but I think discussing the fact that, with a replisome present, there would be no need to stimulate lagging strand synthesis over and above that stimulated by the replisome itself. In lieu of caveats, a stronger argument for the possible role of this activity should be provided.

(6) It isn't clear what the last sentence of the first paragraph in the discussino means:

“The striking similarities among these complexes suggest a general design strategy among complexes containing an internal R-loop”

(Remarks on code availability)

The code is available at the given link, but the README is sparse. It mostly provides a summary of what each code is for, but there did not appear to be instructions of how to use each file.

Reviewer #2

(Remarks to the Author)

In this study, the authors used single-molecule mechanical DNA unzipping to mimic the replication fork action in order to investigate the intrinsic mechanisms as to why RNA Pol II is a more potent roadblock to replication when encountering the fork in head-on orientation. The experimental design is well suited to address the formation of R-loop and the way it promotes the polar roadblock activity of Pol II. By controlling the length of the transcript, the authors provided a strong evidence for the intrinsic polarity of Pol II when the DNA is unzipped in head-on orientation (Figure 1) and unraveled a backtracking behavior of Pol II that can sustain significant unzipping force. Remarkably, this polarity increases with increasing the transcript length and is reduced upon the addition of RNase T1 (Figure 1). Through meticulous experiments, the authors provided a strong evidence for the formation of the RNA-DNA hybrid (Figure 2) and that this process requires the backtracking of Pol II (Figure 3), thus highlighting that the RNA-DNA hybrid is formed in front of Pol II upon head-on collision. Moreover, the authors proposed the formation of a topological lock by the annealing of the RNA to the lagging strand and provided an evidence for their proposition by showing that T7 DNA polymerase can extend the annealed RNA on the lagging strand (Figure 5). Finally, the authors demonstrated potential mechanisms that may remove the RNA-DNA hybrid and allow the backtracked Pol II to reinitiate synthesis, by including TFIS that stimulates the inherent RNA cleavage activity of Pol II and RNase H which is effective in cleaving RNA-DNA hybrid (Figure 4).

Overall, the experiments are well performed and support the conclusions of the study. I have no major concerns to be addressed apart from the two following points below:

Although, the authors presented compelling evidence for the formation of the RNA-DNA topological lock upon the encounter of the mechanically unzipped fork with transcribing Pol II in head-on orientation, extrapolating these findings to the encounter of the replisome with Pol II must be taken with caution. It remains possible that the forefront replisomal proteins,

such as the CMG helicase and the accessory protein FPC, might influence the interactions of Pol II and the DNA-RNA hybrid, therefore reducing the affinity of the Pol II/DNA-RNA complex. For example, if the Pol II interactions with the RNA-DNA hybrid change during DNA unwinding or Pol II backtracking, it is possible that the replisome could sterically slow down and restrict the conformational landscape of these interactions within the Pol II/DNA-RNA complex. An example of such a debate has been heightened in the case of the *E. coli* Tus-Ter roadblock mechanism (DOI: 10.1080/10409238.2017.1394264). On the other hand, RPA coating of the transiently exposed ssDNA on the lagging strand might influence the RNA annealing to the lagging strand. There is no doubt from the evidence presented by the authors that the RNA-DNA topological lock was formed, but to what extent this contributes to the Pol II polar roadblock activity within the contents of the replisome is questionable. The authors could address these questions by 1) comparing the efficiency of their Pol II polar block activity to studies that use the replisome or other motors if they are available; 2) perform the experiment in the presence of helicases; 3) perform the experiment in the presence of RPA. I understand the potential complexity of including helicases but at least the authors should address the suggestions in points 1 and 3.

The results in Figure 1b demonstrated that paused Pol II can backtrack on dsDNA and sustain ~30 pN mechanical unzipping forces for ~100 bp. In addition, in Figure 1d transcribing Pol II can slide on DNA-RNA and sustain similar mechanical unzipping forces but for ~300 bp. Treatment with RNase T1 reduces the sliding distance to those observed in the paused Pol II complex (Figures 1C and 1e). It is interesting that Pol II can backtrack on dsDNA and sustain the mechanical unzipping forces for ~100 bp, hence demonstrating a strong intrinsic interaction with the dsDNA consistent with the structural studies. What is the possibility that this intrinsic high affinity binding of Pol II to dsDNA is sufficient to stall the replisome without the need of the RNA-DNA topological lock under physiological conditions? The Wang's lab reported in previous study the nucleosome ability to induce RNA Pol backtracking and sustain similar mechanical unzipping forces as illustrated in the study (DOI: 10.1038/nsm.1798). I suggest two experiments that might help in addressing this query: 1) to investigate if Pol II display polarity as a roadblock in response to the nucleosome?; 2) to ligate a bubble at the TEC site to prebind Pol II to forked DNA and directly characterize the intrinsic roadblock activity of Pol II.

(Remarks on code availability)

Reviewer #3

(Remarks to the Author)

The manuscript by Kay et al. investigates collisions between RNA polymerase II and DNA replication fork mimics using optical tweezers. The authors explore the influence of collision polarity by initiating transcription either head-on, toward a replication fork, or co-directionally, away from a replication fork. In both cases, force is applied to the ends of the leading and lagging strand arms to mimic mechanical unzipping normally carried out by the helicase. The authors report stronger stability in the head-on configuration that increases with transcript length. They argue that backtracking leads to the formation of an RNA-DNA hybrid structure that forms a topological lock, trapping RNA Pol II at the fork. Addition of TFIIIS, RNase T1 and RNase H alters these observations seemingly providing release pathways. Finally, they show that T7 DNA polymerase can restart synthesis using the RNA-DNA hybrid.

Overall, the manuscript is well written, providing clear explanations for the physical interpretations of the traces, and the figures nicely illustrate features of the very complex experiments. The observation of RNA Pol II backtracking and the formation of RNA-DNA hybrids is truly fascinating and, overall, I find it convincing based on the controls presented. My only significant criticism is the artificial nature of the mechanical unzipping. In my view this simple system is limited because it does not recapitulate the physical encounter involving a DNA replication complex and possible direct interactions. For example, helicases can stall, backtrack, and dissociate. Synthesis by the DNA polymerases might rapidly block any ssDNA available for RNA hybridization. Moreover, it seems unlikely the eukaryotic replication fork will exert the high forces applied in this study. Nevertheless, the beautiful findings presented greatly advance our understanding of RNA Pol II intermediates that may form during DNA replication and will no doubt open up new research directions. The question I raise could have reasonable explanation and still remain open. Therefore, I am supportive of the work. I should also note that it remains unclear whether full reconstitution of eukaryotic transcription and DNA replication at the single molecule level is even possible at this time.

There were, however, controls that I missed and experiments that I could not interpret as a result. Several questions also came up. I have included these in the specific comments below.

Specific Comments:

1. The authors should provide more explanation how they believe the mechanical unzipping of DNA used in the study compares to the actual progression of a replisome. Specifically, it would be helpful to discuss the range of forces expected. How does the force applied during mechanical unzipping compare to the forces generated by a progressing replisome? Could differences in these forces affect the interpretation of the results? (progressing replisome force might not be constant)
2. Looking at the force curves in Figure 1b, I don't understand why the increase in force occurs at different locations of extension. Isn't the RNA Pol II always stalled at the same location? It would help to have a supplemental figure with many transparent traces overlaid (see comment 5) to see if this is a general feature. Have you made a distribution of the start extensions?
3. In the text on page 3 at the bottom, the authors write "suggesting that Pol II can hold on to the DNA fork even without interaction with an RNA-DNA hybrid." This is an intriguing hypothesis. Can the authors test this idea by introducing RNA Pol II in the absence of an RNA primer or bubble. In this case, they are predicting it would bind directly to the fork and increase

the force required to unzipping it. Is this observed?

4. In the experiment where RNase T1 was added in Figure 2, do the authors know that the only influence is removal of long RNA. Have the authors also tried adding RNase T1 to the paused RNA Pol II and determined whether the force to unzip through it is likewise reduced? If true, this would suggest the influence of RNase T1 goes beyond just RNA length and would call for a revision in the interpretation of the experiments.
5. The work would be greatly strengthened by the addition of more example traces for the main conditions to provide a better idea about the diversity of force responses. Also, deposition of all curves in a public database would further enhance the impact by allowing others to learn more about the analysis using the wonderful resources provided by the authors in their Github repository.
6. How do the authors know that TFIS removes the lock by severing the 3' RNA as depicted in Figure 3d? Severing seems like a clear mechanistic explanation for RNase T1 or H, but I wasn't aware of this function for TFIS. Can the authors provide a reference for this?
7. The illustration of backtracking in Figure 3d, really helped to clarify the physical interpretation of backtracking. It would be helpful in there was a little cartoon illustrating the backtracking earlier in the manuscript. I was confused what the authors physically meant when backtracking was first written.
8. What was the concentration of Pol II used in the single-molecule experiments during incubation?
9. How do the authors ensure that Pol II binds only at the transcription start site (TSS)? Do they have a method to remove non-specifically bound Pol II? Have they tested force extension with the DNA substrate lacking a start site after incubation with Pol II? (Related to comment 3)
10. In Figure 2d there are two peaks in the trace. Could this represent two Pol II molecules? Why are two peaks observed?
11. Can the author explain why they switched to RNase H for the later experiments and used RNase T1 in the earlier experiments. It would be helpful to have this information in the main text.
12. No error bar was provided in Figure 4C +RNase H.
13. Is the observation of only 20% hybrid removal in Figure 4c a result of the TFIS concentration? Have the authors tried higher concentrations or do they know this results in full occupancy on Pol II?

(Remarks on code availability)

I quickly looked at the code, but I didn't try running the code myself. I think this is a great resource though. It would be further strengthened by publication of traces that could be used with the code. Without data provided or simulated data provided, it is hard to test the code and evaluate it.

Version 1:

Reviewer comments:

Reviewer #1

(Remarks to the Author)

The authors have done a considerable number of new experiments to improve the manuscript and have responded in detail to comments in the first review. Quite impressive!

I would recommend acceptance given attention to the below two suggestions:

(1) I have a lingering concern with respect to the interpretation of the RPA experiments presented in Supplementary Figure 7. One might expect that the presence of RPA alone (and the formation of an RPA filament on the regions of ssDNA) should act to prevent DNA rewinding. In fact, the authors show in SF7c that RPA does in fact prevent ssDNA-ssDNA interactions. If so, it would seem that one can't interpret the lack of rewinding in the presence of RPA as being due to an RNA-DNA hybrid and can't conclude that, "that hybrid formation effectively competes with RPA binding to ssDNA".

However, the control group of paused A20 polymerases, which lack the RNA to make an RNA-DNA hybrid do show that it is indeed the RNA that is required as opposed to an effect of RPA itself. This control should be pointed out in the text and used to more strongly support the conclusion.

Lastly, since there are differences between the paused and elongating polymerases, a further control (if the authors were so inclined) would be the inclusion of RNase T1 in the presence of RPA. The expectation is that one would lose the blockage of re-zipping under these conditions as is seen in the absence of RPA.

(2) I also suggest statistical analysis of the +/- RNase T1 distributions presented in Figure 1c and Supplementary Figure 4. Under which conditions are the distributions consistent with an effect?

(Remarks on code availability)

Reviewer #2

(Remarks to the Author)

The authors adequately address my two major comments through performing the experiment in the presence of RPA, conducting additional analysis and introducing changes to the text. The revised manuscript is significantly improved. I have no further comments.

(Remarks on code availability)

There is no code, but the authors provided the raw data for the figures in Github for those who are interested in reanalyzing the data, which I think is sufficient to address any future query.

Reviewer #3

(Remarks to the Author)

The authors have done an excellent job addressing my comments. They performed additional experiments, analysis, and provided additional figures. Specifically, they now demonstrate that RNase T1 does not impact any detectable properties of the elongation complex. They also demonstrate T7 RNAP has similar polar roadblock results as RNAPII. Finally, they were not able to demonstrate fully that mechanically unzipping DNA is entirely similar to the replisome. However, they have modified the text to accurately reflect the limitations for their approach.

This paper is ready for publication. I have no further comments.

(Remarks on code availability)

Response to Reviewers

Re: Nature Communications manuscript NCOMMS-24-80678-T

RNA Polymerase II is a Polar Roadblock to a Progressing DNA Fork by Kay et al.

We greatly appreciate the helpful comments and suggestions from the three Reviewers. We have carefully considered each comment and taken comprehensive actions to address them. These comments prompted us to conduct three new experiments. The results of these experiments have provided new insights and further strengthened the original conclusions.

Below, we start with a summary of major changes to the manuscript. This is followed by a detailed, point-by-point response to each comment from the three Reviewers, with each comment (bold) followed by our response (not bold). In the revised manuscript, we have marked the suggested changes in red.

Summary of Major Changes:

1. Reviewer #1 and Reviewer #2 pointed out the possibility that single-stranded binding protein RPA may inhibit RNA-DNA hybrid formation during a head-on collision of Pol II with a DNA fork. We have conducted new experiments in the presence of RPA and found that the hybrid can still form relatively efficiently. These results broaden the significance of our work and are now included as a new SI figure, Supplementary Fig. 7.
2. Reviewer #3 asked us to conduct polarity experiments with Pol II paused at A20 in both co-directional and head-on orientations in the presence of RNase T1. We have performed these experiments. These new results, which are incorporated into Fig. 1c, further demonstrate the role of RNA in enhancing the disruption force and sliding distance, consistent with the results obtained by elongating Pol II in the presence of RNase T1.
3. Reviewer #2 and Reviewer #3 inquired about the nature of Pol II sliding in the head-on orientation, especially when Pol II is initially paused at A20. We re-analyzed these data to show the sliding time of this complex. Our data suggest that the longer distance sliding requires some length of nascent RNA and assumes a configuration with a lifetime of a few seconds. This new analysis and discussion are now provided as a new SI figure, Supplementary Fig. 4.
4. Reviewer #3 inquired if RNase T1 can influence Pol II activities beyond RNA cleavage. To more definitively answer this question, we have conducted another head-on orientation experiment using an assembled elongation complex with an RNA that cannot be cleaved by RNase T1, both in the presence and absence of RNase T1. These new results suggest that RNase T1 does not influence any detectable properties of the elongation complex. They are now shown as a new SI figure, Supplementary Fig. 4.
5. Reviewer #2 asked us to compare the efficiency of their Pol II polar block activity with that of other motors, if they are available. To the best of our knowledge, we are not aware of any prior studies, except for our own studies of *E. coli* RNA polymerase using the DNA unzipping mapper method in both co-directional and head-on orientations (DOI: 10.1016/j.cell.2017.11.017). Our re-analysis of those data shows that *E. coli* RNAP exhibits a similar polarity to the Pol II roadblock at a DNA fork,

suggesting that the polarity may be the same for the transcription elongation complex in both prokaryotes and eukaryotes. This new analysis is now shown as a new SI figure, Supplementary Fig. 9.

6. Reviewer 3 asked us to show more example traces and inquired about the positional homogeneity of the elongation complex. We have added a new Supplementary Figure, Supplementary Fig. 3 of the revised manuscript, to show an overlay of additional representative traces and a histogram of Pol II location on the DNA sequence. This analysis shows that, although Pol II starts from a relatively homogeneous position, the sliding behaviors differ significantly from trace to trace.

Reviewer #1 (Remarks to the Author):

Review of NatComm RNAP against unzipping block

This work from a leader in the field of single-molecule based investigations of protein-nucleic acid systems describes interesting experiments aimed at understanding what happens when an RNA polymerase encounters a strand-separated fork. They probe different directionalities of the encounter with respect to the direction of transcription and show that RNA-DNA hybrids may form between the nascent transcript and one or the other of the ssDNAs in the fork. They also show that these hybrids increase the block to fork progression; especially one formed in front of a backtracked RNAP. As expected, TFIIS and RNase T1 release this extra blockage by cleaving the RNA. Lastly, they show that this RNA-DNA hybrid may serve as a primer for T7 DNA polymerase.

The experiments are very well done and well controlled. The paper is also clearly written, and it is straightforward to follow the logic presented. The results clearly augment our understanding of the molecular outcomes of the studied system.

We very much appreciate these encouraging comments from the Reviewer.

My only concerns are based on what I feel is a bit of stretching as to the significance and interpretation of the data which I detail below:

(1) As the authors themselves introduce, we already know about the different outcomes of head-on and trailing collisions. In fact, 10 references are provided which describe this. Thus, what is the actual gap in knowledge that is being pursued in this work?

The authors claim that they are after a “fundamental mechanism” underlying this effect of directionality. However, when rationalizing why a head-on is different than a trailing collision, they restate observations made from previous structural studies dealing with the strength of upstream and downstream interactions between the polymerase and the DNA. This mechanism could have been done without any of the single-molecule data. At the end of page 4, they write: “Based on the Pol II EC crystal structure and supported by these data, Pol II interacts weakly with dsDNA behind its active site but tightly with dsDNA in front of its active site. These structural features provide a possible explanation for the Pol II polarity detected by the DNA fork and might also hold for a replisome.”

In my opinion, the initial single-molecule data simply confirms previous observations which takes away a bit of its significance.

In contrast, the observation that seems a bona fide new piece of information revealed by this specific approach is the topological lock that may form with the head-on collision, backtracking RNAP.

We very much appreciate this comment from the Reviewer. Upon reading this comment, we realized that we had not accurately discussed the crucial role our work plays in understanding the nature of polarity. Although the existing elongation complex structure provides some context for our work, it is not sufficient to explain this polarity.

To remove the Pol II roadblock in either direction, the fork must progress through the elongation complex, disrupting interactions located both upstream and downstream of Pol II's active site. If the Pol II roadblock strength solely depends on the sum of these interactions, then the Pol II roadblock should have no polarity to a progressing fork. Instead, our findings suggest that the polarity depends on how the fork approaches an elongation complex. In the co-directional orientation, the DNA fork first encounters the upstream edge of the transcription bubble as indicated by the structure of the elongation complex. The low disruption force from our work suggests that after the fork separates the two DNA strands in the bubble, Pol II's DNA clamp, downstream of the active site, becomes weaker, rendering the elongation complex unstable. In contrast, in the head-on orientation, the fork first encounters the front edge of Pol II downstream of its active site. The high disruption force from our work suggests that Pol II can tightly clamp onto the DNA downstream of its active site, strongly resisting destabilization by an advancing fork.

We hope that this discussion better clarifies what is known based on the structural studies and what is revealed through our data. This revised discussion is now provided on page 5 of the revised manuscript.

(2) Is a mechanically unwound fork a good fill in for a replisome? It seems taken for granted that the experimental setup reasonably mimicks a replisome. However, there are almost certainly caveats to this. In the context of a replisome, with the leading replicative helicase out in front, the accessible ssDNA would be significantly farther away from the polymerase itself and would not interact directly. The actual interaction would probably be mediated via protein:protein interactions between the helicase and the polymerase. To this point, the authors themselves comment in the text (again at the end of page 4) that:

“Since the progressing DNA fork used here somewhat resembles a progressing replication fork, our observations might provide a mechanistic explanation for previous findings that head-on transcription-replication conflicts are more detrimental than co-directional ones.”

Please provide clarification as to what “somewhat resembles” means. It sounds like a pretty strong caveat that merits discussion.

We understand this concern. Although the DNA fork may reasonably mimic the replisome, we agree with the Reviewer that a complete understanding of RNA-DNA hybrid formation during a transcription-replication conflict requires the full reconstitution of both machineries, an experimental feat that has yet to be demonstrated biochemically with eukaryotic systems. The progressing DNA fork used here somewhat resembles a progressing replication fork in that the DNA fork is mechanically unwound rather than being unwound by CMG. It is possible that such a simple system does not fully capture the

intricacies of a replisome, which involves interactions among multiple proteins. Nonetheless, our findings might still provide some mechanistic insights into the polarity of the Pol II roadblock during transcription-replication conflicts.

We have now commented on this on page 2 of the revised manuscript.

(3) As an add on to the previous point, it seems unlikely that in a replisome the ssDNA strands are splayed out at 180 degrees from one another. This geometry artificially brings the ssDNA closer to the polymerase. I wonder what the effect of holding the strands a different (more acute) angle would have with the prevalence of RNA-DNA hybrid formation. In principle, this could be done with the tweezers, but I understand it may represent a considerably amount of work.

We appreciate the Reviewer's comment regarding the DNA fork geometry. Our experimental configuration pulls the leading and lagging strands in opposite directions, which may not fully reflect the strand geometry within a replisome. It is worth noting that since the ssDNA is highly flexible, with a persistence length of only 1.5 nt, the ssDNA near the fork should be able to adopt a broad range of configurations in solution, although the replisome structure may capture a specific angle.

It is unclear whether our experimental configuration favors or disfavors the formation of an RNA-DNA hybrid in front of Pol II during a head-on collision with a fork. To form a hybrid, ssDNA and RNA must explore different conformational spaces to nucleate a hybrid. Nucleation may occur more readily when both RNA and ssDNA are relaxed, allowing them to explore various configurations more freely. On the other hand, relaxed ssDNA can adopt secondary structures that may inhibit base pairing with RNA. Our experimental configuration places the ssDNA under tension, restricting its conformational space, which may make hybrid formation more difficult. On the other hand, this presence of force will also prevent the formation of secondary structures in the ssDNA (as the Reviewer noted below), which will favor hybrid formation.

As the Reviewer noted, addressing this problem requires a systematic examination of how different fork geometries impact the formation of RNA-DNA hybrids. Although it falls beyond the scope of our current work, it is an interesting topic that we would like to explore in our future research.

(4) As a second add on to the degree to which the splayed fork resembles a replisome, an actively replicating system is known to have the ssDNA coated with single-stranded DNA binding proteins. It would seem straightforward that SSBs would inhibit the formation of the RNA-DNA hybrids revealed here.

Relatedly, even in the absence of SSBs, without the ssDNA held under tension, it would have the propensity to form secondary structures which may also prevent (or decrease) the formation of RNA-DNA hybrids.

The comment, regarding ssDNA secondary structures, is related to the one above. Please also see our response there.

The Reviewer commented that the presence of the single-stranded DNA binding protein, such as RPA, may inhibit the formation of RNA-DNA hybrids, because RPA binds to ssDNA and can thus compete with RNA for binding. Inspired by this comment, we conducted new experiments similar to those shown in

Fig. 3 but with 5 nM RPA present during transcription. This concentration is more than an order of magnitude higher than the K_d of RPA to ssDNA (DOI: 10.1128/mcb.12.7.3050-3059.1992).

We found that RPA binding to ssDNA can effectively resist DNA re-zipping. Without RPA, DNA can be re-zipped at around 15 pN; however, with RPA, re-zipping is prevented until the force is lowered to around 5 pN, at which point re-zipping becomes energetically favorable and the DNA becomes fully re-zipped. Furthermore, as the Reviewer has already pointed out, without RPA, ssDNA forms secondary structures in the low force regime. Consistent with this, the measured force exceeds that predicted in the low-force regime. In the presence of RPA, we found that RPA inhibits the formation of ssDNA secondary structures in the low force regime and significantly lowers the force in this regime. Importantly, in the presence of RPA, an RNA-DNA hybrid can still form relatively efficiently, indicating that hybrid formation can effectively compete with RPA binding to ssDNA.

These new results are now presented as a new SI figure, Supplementary Fig. 7.

(5) The data convincingly show that an RNA primer enables polymerization. Very cool experiment, but as the replisome already needs to exist in order to collide with the polymerase, its not clear that this behavior would ever contribute to replication. This experiment seems more confidently interpreted as another control that in fact, the RNA-DNA hybrid is forming where the authors say it is rather than a regulatory or biologically important point.

The authors do say that this is a proof of principle only due to the use of T7 instead of pol delta, but I think discussing the fact that, with a replisome present, there would be no need to stimulate lagging strand synthesis over and above that stimulated by the replisome itself. In lieu of caveats, a stronger argument for the possible role of this activity should be provided.

We appreciate this comment from the Reviewer. Following the Reviewer's comment, under the Results section on "RNA-DNA hybrid enables lagging-strand replication", we have removed the introductory sentences related to *in vivo* relevance. Instead, we motivated this experiment as further validation of the formation of the RNA-DNA hybrid. At the end of this section, we point out the possibility that in eukaryotes, such a hybrid can be used for lagging strand synthesis, since the RNA primer is already laid down. This is simply stated as a possibility. Furthermore, we have removed any discussion of these results from the Discussion section.

(6) It isn't clear what the last sentence of the first paragraph in the discussino means:

"The striking similarities among these complexes suggest a general design strategy among complexes containing an internal R-loop"

We agree that this sentence can be confusing. We have revised the corresponding paragraph to focus the discussion on the transcription elongation complex on page 10 of the revised manuscript.

Reviewer #1 (Remarks on code availability):

The code is available at the given link, but the README is sparse. It mostly provides a summary of what each code is for, but there did not appear to be instructions of how to use each file.

We have revised the "read me" file under the GitHub link to provide more description for each code.

Reviewer #2 (Remarks to the Author):

In this study, the authors used single-molecule mechanical DNA unzipping to mimic the replication fork action in order to investigate the intrinsic mechanisms as to why RNA Pol II is a more potent roadblock to replication when encountering the fork in head-on orientation. The experimental design is well suited to address the formation of R-loop and the way it promotes the polar roadblock activity of Pol II. By controlling the length of the transcript, the authors provided a strong evidence for the intrinsic polarity of Pol II when the DNA is unzipped in head-on orientation (Figure 1) and unraveled a backtracking behavior of Pol II that can sustain significant unzipping force. Remarkably, this polarity increases with increasing the transcript length and is reduced upon the addition of RNase T1 (Figure 1). Through meticulous experiments, the authors provided a strong evidence for the formation of the RNA-DNA hybrid (Figure 2) and that this process requires the backtracking of Pol II (Figure 3), thus highlighting that the RNA-DNA hybrid is formed in front of Pol II upon head-on collision. Moreover, the authors proposed the formation of a topological lock by the annealing of the RNA to the lagging strand and provided an evidence for their proposition by showing that T7 DNA polymerase can extend the annealed RNA on the lagging strand (Figure 5). Finally, the authors demonstrated potential mechanisms that may remove the RNA-DNA hybrid and allow the backtracked Pol II to reinitiate synthesis, by including TFIS that stimulates the inherent RNA cleavage activity of Pol II and RNase H which is effective in cleaving RNA-DNA hybrid (Figure 4).

Overall, the experiments are well performed and support the conclusions of the study. I have no major concerns to be addressed apart from the two following points below:

We very much appreciate these positive remarks from the Reviewer.

Although, the authors presented compelling evidence for the formation of the RNA-DNA topological lock upon the encounter of the mechanically unzipped fork with transcribing Pol II in head-on orientation, extrapolating these findings to the encounter of the replisome with Pol II must be taken with caution. It remains possible that the forefront replisomal proteins, such as the CMG helicase and the accessory protein FPC, might influence the interactions of Pol II and the DNA-RNA hybrid, therefore reducing the affinity of the Pol II/DNA-RNA complex. For example, if the Pol II interactions with the RNA-DNA hybrid change during DNA unwinding or Pol II backtracking, it is possible that the replisome could sterically slow down and restrict the conformational landscape of these interactions within the Pol II/DNA-RNA complex. An example of such a debate has been heightened in the case of the *E. coli* Tus-Ter roadblock mechanism (DOI: 10.1080/10409238.2017.1394264). On the other hand, RPA coating of the transiently exposed ssDNA on the lagging strand might influence the RNA annealing to the lagging strand. There is no doubt from the evidence presented by the authors that the RNA-DNA topological lock was formed, but to what extent this contributes to the Pol II polar roadblock activity within the contents of the replisome is questionable. The authors could address these questions by 1) comparing the efficiency of their Pol II polar block activity to studies that use the replisome or other motors if they are available; 2) perform the experiment in the presence of helicases; 3) perform the experiment in the presence of RPA. I understand the potential complexity of including helicases, but the authors should at least address the suggestions in points 1 and 3.

We agree with the Reviewer that the interpretation of our results must be taken with caution, as our system uses the DNA fork to mimic the eukaryotic replication machinery and will not be able to fully capture the complexity of the replisome and its interaction with Pol II. We have now commented on this on page 2 of the revised manuscript. In this revision, we have also removed most of the discussion related to the potential *in vivo* use of the RNA-DNA hybrid for lagging strand replication, as suggested by Reviewer #1.

We understand the comment of this Reviewer that experiments presented in our initial manuscript were performed in the absence of RPA, and the presence of RPA may inhibit the ability of RNA to hybridize with the lagging strand to form an RNA-DNA hybrid, because RPA binds to ssDNA and can thus compete with RNA for binding. This comment was also raised by Reviewer #1.

Inspired by this comment, we conducted new experiments similar to those shown in Fig. 3 but with 5 nM RPA present during transcription. This concentration is more than an order of magnitude higher than the K_d of RPA to ssDNA (DOI: 10.1128/mcb.12.7.3050-3059.1992). We found that RPA binding to ssDNA can effectively resist DNA re-zipping. Without RPA, DNA can be re-zipped at around 15 pN. However, with RPA, re-zipping is prevented until the force is lowered to around 5 pN, at which point re-zipping becomes energetically favorable and the DNA becomes fully re-zipped. In addition, without RPA, ssDNA forms secondary structures in the low-force regime, resulting in a force in the ssDNA force-extension curve that is greater than predicted. With RPA, however, RPA inhibits the formation of ssDNA secondary structures, significantly lowering the force in the low-force regime. Importantly, in the presence of RPA, an RNA-DNA hybrid can still form relatively efficiently, indicating that hybrid formation can effectively compete with RPA binding to ssDNA. These new results are now presented as a new SI figure, Fig. 7.

The Reviewer also suggested that we compare the Pol II polar block activity with that of other motors. To the best of our knowledge, there have been no such reports of such prior studies, except for our studies of *E. coli* RNA polymerase (RNAP) using the DNA unzipping mapper method in both co-directional and head-on orientations (DOI: 10.1016/j.cell.2017.11.017). We reanalyzed the data and plotted the disruption force. We found that *E. coli* RNAP exhibits a similar polarity to the Pol II roadblock at a DNA fork, displaying a greater disruption force in the head-on orientation than in the co-directional orientation. This suggests that the polarity may be the same for the transcription elongation complex in both prokaryotes and eukaryotes. This new analysis is now shown as a new SI figure, Supplementary Fig. 9. We have also commented on this on page 10 of the revised manuscript.

The Reviewer also suggested that we perform the experiments in the presence of helicase. The appropriate helicase would be eukaryotic CMG. This is an excellent experiment that we would be very interested in exploring. It is, unfortunately, outside the scope of the current work.

The results in Figure 1b demonstrated that paused Pol II can backtrack on dsDNA and sustain ~30 pN mechanical unzipping forces for ~100 bp. In addition, in Figure 1d transcribing Pol II can slide on DNA-RNA and sustain similar mechanical unzipping forces but for ~300 bp. Treatment with RNase T1 reduces the sliding distance to those observed in the paused Pol II complex (Figures 1C and 1e). It is interesting that Pol II can backtrack on dsDNA and sustain the mechanical unzipping forces for ~100 bp, hence demonstrating a strong intrinsic interaction with the dsDNA consistent with the structural studies. What is the possibility that this intrinsic high affinity binding of Pol II to dsDNA is sufficient to stall the replisome without the need of the RNA-DNA topological lock under physiological conditions? The Wang's lab reported in previous study the nucleosome ability to induce RNA Pol backtracking and

sustain similar mechanical unzipping forces as illustrated in the study (DOI: 10.1038/nsmb.1798). I suggest two experiments that might help in addressing this query: 1) to investigate if Pol II display polarity as a roadblock in response to the nucleosome?; 2) to ligate a bubble at the TEC site to prebind Pol II to forked DNA and directly characterize the intrinsic roadblock activity of Pol II.

Just as the Reviewer, we also found it interesting that Pol II can slide backwards over a long distance in the head-on orientation. This is particularly true for the A20 paused Pol II, which only has an RNA of 25 nt, while being able to slide over a distance much greater than the RNA length. For a sliding distance smaller than the transcript length, Pol II may achieve this reverse motion via backtracking, where the transcription bubble maintains an RNA-DNA hybrid within the bound Pol II. However, if the sliding distance exceeds the transcript size, the RNA-DNA hybrid can no longer be maintained within the transcription bubble. For this case, it is unclear how Pol II remains bound to the DNA. One possibility is for Pol II to assume a configuration akin to an open complex, which is known to be short-lived and prone to bubble collapse over a time scale of seconds (DOI: 10.1038/nsmb1280). This state is very similar to what the Reviewer suggested, namely, ligating a bubble at the TEC site to prebind Pol II to a forked DNA, effectively forming an open complex of transcription. Although it is challenging to create such a complex due to its limited stability, our experimental method naturally lends itself to investigating the possibility that the long-distance sliding of the A20-paused Pol II might be a result of the sliding of an open complex. As shown in the new SI figure, Supplementary Fig. 4, we examined the sliding time of an A20-paused Pol II in the head-on orientation. The lifetime of this configuration is short (a few seconds), similar to the lifetime of an open complex. However, the paused A20 Pol II data also suggest that long-distance sliding increases with RNA transcript size, as the presence of RNase T1 reduces the sliding distance. This indicates that the sliding configuration may not be a simple open complex, but rather one that also involves RNA, with its ability to slide likely dependent on the DNA sequence. Thus, this configuration is short-lived, but the exact nature of the orientation remains unclear.

When we unzip naked DNA (without any pre-formed elongation complex) and then introduce Pol II to the sample chamber, we do not detect any force that rises significantly above the naked DNA baseline. This indicates that binding of Pol II to dsDNA without a transcription bubble containing an RNA-DNA hybrid is unstable, so Pol II alone does not have any intrinsic high-affinity to dsDNA. Bona fide Pol II transcription initiation requires assembly of the preinitiation complex, with general transcription factors bound at the promoter.

The Reviewer also suggested that we examine whether Pol II displays polarity as a roadblock in response to the nucleosome. A nucleosome is not a motor, so it remains relatively stationary and cannot actively approach Pol II. It always encounters Pol II from the front of Pol II when Pol II encounters it during transcription. It can only encounter Pol II from the back of Pol II if Pol II backtracks, or if nucleosome remodeling enzymes actively relocate the nucleosome. This problem may be interesting to explore, but it is beyond the scope of the current work.

Reviewer #3 (Remarks to the Author):

The manuscript by Kay et al. investigates collisions between RNA polymerase II and DNA replication fork mimics using optical tweezers. The authors explore the influence of collision polarity by initiating transcription either head-on, toward a replication fork, or co-directionally, away from a replication fork. In both cases, force is applied to the ends of the leading and lagging strand arms to mimic mechanical unzipping normally carried out by the helicase. The authors report stronger stability in the

head-on configuration that increases with transcript length. They argue that backtracking leads to the formation of an RNA-DNA hybrid structure that forms a topological lock, trapping RNA Pol II at the fork. Addition of TFIIIS, RNase T1 and RNase H alters these observations seemingly providing release pathways. Finally, they show that T7 DNA polymerase can restart synthesis using the RNA-DNA hybrid.

Overall, the manuscript is well written, providing clear explanations for the physical interpretations of the traces, and the figures nicely illustrate features of the very complex experiments. The observation of RNA Pol II backtracking and the formation of RNA-DNA hybrids is truly fascinating and, overall, I find it convincing based on the controls presented. My only significant criticism is the artificial nature of the mechanical unzipping. In my view this simple system is limited because it does not recapitulate the physical encounter involving a DNA replication complex and possible direct interactions. For example, helicases can stall, backtrack, and dissociate. Synthesis by the DNA polymerases might rapidly block any ssDNA available for RNA hybridization. Moreover, it seems unlikely the eukaryotic replication fork will exert the high forces applied in this study. Nevertheless, the beautiful findings presented greatly advance our understanding of RNA Pol II intermediates that may form during DNA replication and will no doubt open up new research directions. The question I raise could have reasonable explanation and still remain open. Therefore, I am supportive of the work. I should also note that it remains unclear whether full reconstitution of eukaryotic transcription and DNA replication at the single molecule level is even possible at this time.

We appreciate the Reviewer for recognizing the complexity of these measurements and the value of using a simple system to gain mechanistic insights into a highly complicated process. We also agree with the Reviewer that our mechanical unwinding of the DNA fork cannot fully recapitulate many aspects of an advancing replisome and possible direct interactions of the replisome with Pol II. A complete understanding of this process will indeed require the full reconstitution of eukaryotic transcription and replisome, which is yet to be demonstrated biochemically with a eukaryotic system. We have now commented on this on page 2 of the revised manuscript.

There were, however, controls that I missed and experiments that I could not interpret as a result. Several questions also came up. I have included these in the specific comments below.

Specific Comments:

1. The authors should provide more explanation how they believe the mechanical unzipping of DNA used in the study compares to the actual progression of a replisome. Specifically, it would be helpful to discuss the range of forces expected. How does the force applied during mechanical unzipping compare to the forces generated by a progressing replisome? Could differences in these forces affect the interpretation of the results? (progressing replisome force might not be constant)

We appreciate this question regarding the force a progressing replisome can generate. This is a fascinating question. To the best of our knowledge, this force has never been determined. Importantly, this force needs to be discussed in the context of a head-on transcription-replication conflict: the replisome forward translocation unwinds (unzips) the DNA, while Pol II forward translocation rewinds (rezip) the DNA. The observation that the replisome often wins this conflict suggests that the replisome can generate a stronger unzipping force than the reziping force of Pol II. Although our lab and others have measured the forces DNA-based motors can exert by pulling on DNA, there are limited studies of the forces a motor generates to rezip the DNA fork. The only published work is our prior studies of *E. coli*

RNAP (DOI: 10.1016/j.cell.2017.11.017). In that work, we show that *E. coli* RNAP can generate 19 pN of force to rezip the DNA by itself and 22 pN of force with the help of Mfd. Assuming that Pol II can generate similar forces on its own and with other factors, then the eukaryotic replisome should be able to generate a force greater than 22 pN. In our work, the DNA fork exerts a force of about 35 pN to disrupt Pol II (Fig. 1).

2. Looking at the force curves in Figure 1b, I don't understand why the increase in force occurs at different locations of extension. Isn't the RNA Pol II always stalled at the same location? It would help to have a supplemental figure with many transparent traces overlaid (see comment 5) to see if this is a general feature. Have you made a distribution of the start extensions?

In Fig. 1b, Pol II paused as A20 is indeed stalled at a defined location along the DNA sequence. However, the DNA templates for the co-directional and head-on orientations require different designs, because the DNA fork must approach the paused Pol II from two different orientations. For the co-directional template, the initial fork position is located around 700 bp from the paused Pol II, whereas for the head-on template, the initial fork position is located 1600 bp from the paused Pol II. Please see Supplementary Figure 2, which provides details of the template designs.

In Fig. 1b, the RNA is 25 nt and thus can backtrack a small distance. What is surprising is that Pol II can slide a distance much greater than the RNA length, such that it no longer maintains an elongation complex. We do not know the exact structure of such a long-distance sliding complex. It is possible that Pol II assumes a configuration of an open complex. During the sliding, the force variations may reflect the energetics of complex formation at different locations along the DNA, which could include re-zipping the DNA at the front of the transcription bubble, unzipping the DNA at the back of the transcription bubble, the stability of the RNA-DNA hybrid (if any), and the affinities of Pol II to clamp down onto the DNA downstream of its active site, all of which can also depend on the DNA sequence and will result in different force values. Our new SI figure, Supplementary Fig. 4 of the revised manuscript, also suggests a more complex configuration that may not be a simple open complex but one that also involves RNA.

We have followed the suggestion from this Reviewer and added a new Supplementary Figure, Fig. 3 of the revised manuscript. In this figure, we show an overlay of five additional representative traces. Additionally, we show a histogram of Pol II location on the DNA sequence, detected from a force rise in the force-extension curve. This histogram indicates that the starting position of Pol II is relatively homogeneous, with a mean close to the expectation and a STD of 8 bp. Yet, the sliding behaviors are very different.

3. In the text on page 3 at the bottom, the authors write "suggesting that Pol II can hold on to the DNA fork even without interaction with an RNA-DNA hybrid." This is an intriguing hypothesis. Can the authors test this idea by introducing RNA Pol II in the absence of an RNA primer or bubble. In this case, they are predicting it would bind directly to the fork and increase the force required to unzipping it. Is this observed?

This comment is related to the comment above and a comment from Reviewer #2. Just as the Reviewer, we also found it intriguing that the A20 paused Pol II, which only has an RNA of 25 nt, can slide over a distance much greater than the RNA length. It is unclear how Pol II remains bound to the DNA without an RNA-DNA hybrid within the complex. One possibility is for Pol II to assume an open complex, which is known to be short-lived and prone to bubble collapse over a time scale of seconds (DOI: 10.1038/nsmb1280). As shown in the new SI figure, Supplementary Fig. 4, we examined the sliding time

of the A20 paused Pol II and found a lifetime of a few seconds, similar to the lifetime of an open complex. However, the paused A20 Pol II data also suggest that long-distance sliding increases with RNA transcript size, as the presence of RNase T1 reduces the sliding distance. This indicates that the sliding configuration may not be a simple open complex, but rather one that also involves RNA, with its ability to slide likely dependent on the DNA sequence. Thus, this configuration is short-lived, but the exact nature of the orientation remains unclear.

The Reviewer raises another possible sliding configuration – Pol II binding to DNA in the absence of an RNA primer or bubble. Previous single-molecule studies show that such a complex is highly unstable with a lifetime of ~ 30 ms for *E. coli* RNAP (DOI: 10.1038/nsmb.2472) and ~145 ms for Pol II (DOI: 10.1038/nsmb1280). Consistent with this, when we unzip naked DNA (without any pre-formed elongation complex) and then introduce Pol II to the sample chamber, we did not detect any force rise significantly above the naked DNA baseline. We do not expect such a complex to be capable of resisting the progression of the DNA fork. The resistance requires Pol II to bind to DNA in a more stable configuration.

4. In the experiment where RNase T1 was added in Figure 2, do the authors know that the only influence is removal of long RNA. Have the authors also tried adding RNase T1 to the paused RNA Pol II and determined whether the force to unzipp through it is likewise reduced? If true, this would suggest the influence of RNase T1 goes beyond just RNA length and would call for a revision in the interpretation of the experiments.

Following the suggestion by this Reviewer, we conducted polarity experiments with Pol II paused at A20 in both co-directional and head-on orientations in the presence of RNase T1. Please note that the paused A20 Pol II complex contains a 25 nt RNA, so that RNA outside the elongation complex can potentially be cleaved by RNase T1. We found that the presence of RNase T1 slightly decreases both the disruption force and the sliding distance. These new results are now presented in Fig. 1c of the revised manuscript. These findings are consistent with those from the elongating Pol II, which demonstrate a more significant reduction in disruption force and sliding distance, which we attribute to its longer RNA.

To more definitively address this question from the Reviewer. We conducted an additional head-on orientation experiment using the assembled elongation complex before any NTP chase. This complex contains 14 nt RNA, with 9 nt RNA complementary to the template strand and 5' UUUUU tail non-complementary to the template (see Methods). The U-tail can potentially stabilize the elongation complex via interactions with the RNA channel of Pol II (DOI: 10.4161/trns.20269), but it cannot be cut by RNase T1. We found that the disruption force and sliding distance remain the same in the absence and presence of RNase T1, indicating that RNase T1 does not impact any detectable properties of the elongation complex. These results are now shown as a new SI figure, Supplementary Fig. 4.

5. The work would be greatly strengthened by the addition of more example traces for the main conditions to provide a better idea about the diversity of force responses. Also, deposition of all curves in a public database would further enhance the impact by allowing others to learn more about the analysis using the wonderful resources provided by the authors in their Github repository.

This comment is related to Comment 2 of this Reviewer, so please see our response there. We have shown an overlay of additional representative head-on traces in a new supplementary figure, Supplementary Fig. 3.

In addition, we have included a Source Data file with this revision. This file contains all data shown in both the main and supplementary figures. Thus, the readers should have full access to both the code and its input data.

6. How do the authors know that TFIS removes the lock by severing the 3' RNA as depicted in Figure 3d? Severing seems like a clear mechanistic explanation for RNase T1 or H, but I wasn't aware of this function for TFIS. Can the authors provide a reference for this?

When RNA Pol II is backtracked, the 3' end of the RNA transcript becomes misaligned from the active site, halting transcription. TFIS binds to the secondary channel of Pol II and is known to stimulate the intrinsic RNA cleavage activity of Pol II, generating a new 3' end aligned with the active site, allowing elongation to resume. Many prior studies have documented this behavior of TFIS. Please see our citations on page 8 of the revised manuscript. Here is an early reference from Caroline Kane (DOI: 10.1073/pnas.91.17.8087).

7. The illustration of backtracking in Figure 3d, really helped to clarify the physical interpretation of backtracking. It would be helpful if there was a little cartoon illustrating the backtracking earlier in the manuscript. I was confused what the authors physically meant when backtracking was first written.

When we discuss Fig. 1 results, we use the term "slide backwards", which refers to a generic behavior of any reverse motion. When discussing the results in Fig. 3, we introduce the term "backtracking," which is a subset of sliding backwards. Perhaps "slide backwards" can be confused with "backtracking", which has a specific definition in the transcription field. The cartoon shown in Fig. 3d shows the configuration of a backtracked complex.

8. What was the concentration of Pol II used in the single-molecule experiments during incubation?

Pol II was pre-assembled onto DNA into a transcription elongation complex. Thus, the concentration of Pol II is the same as the DNA concentration used in the experiments: 2.5 pM. This is now indicated on page 15 of the revised manuscript.

9. How do the authors ensure that Pol II binds only at the transcription start site (TSS)? Do they have a method to remove non-specifically bound Pol II? Have they tested force extension with the DNA substrate lacking a start site after incubation with Pol II? (Related to comment 3)

Since we assemble the transcription elongation complex (see Methods), the transcription initiation step is bypassed. That is, the transcription start site (TSS) is defined by the RNA used during the assembly. This method is highly specific, placing an active Pol II at a particular location on DNA. We do not detect any force rise until the DNA fork encounters a bound Pol II at the expected location, indicating minimal non-specific Pol II binding to DNA. Please also see our response to comment 3 above.

10. In Figure 2d there are two peaks in the trace. Could this represent two Pol II molecules? Why are two peaks observed?

Fig. 2d actually shows a cluster of force rise, which indicates sliding. As we noted above, it is impossible for two Pol II molecules to bind to the same DNA molecule, given our method of elongation complex assembly. The force cluster indicates a single Pol II molecule sliding over a distance; for this trace, Pol II

has transcribed 596 bp. Note that one mode of sliding is backtracking, but sliding can also take on other form.

11. Can the author explain why they switched to RNase H for the later experiments and used RNase T1 in the earlier experiments. It would be helpful to have this information in the main text.

RNase T1 cleaves RNA, so it is more suited for our earlier experiments to truncate the RNA size. RNase H can cleave the RNA that is associated with an RNA-DNA hybrid. This is now indicated on page 8 of the revised manuscript.

12. No error bar was provided in Figure 4C +RNase H.

The error bar for it is zero since 100% of the traces out of 20 show hybrid removal.

13. Is the observation of only 20% hybrid removal in Figure 4c a result of the TFIS concentration? Have the authors tried higher concentrations or do they know this results in full occupancy on Pol II?

We used a concentration of TFIS of 1 μ M, which is considered a saturating condition since the K_d of TFIS binding to an elongation complex has been previously measured to be 275 nM (DOI: 10.1073/pnas.0409405102). This citation also shows that the TFIS cleavage rate is slow. In our experiment, the Pol II backtracked complex is present approximately 35 s, which may not be a sufficiently long time for TFIS to perform its function. This could explain why hybrid removal is not 100%.

Reviewer #3 (Remarks on code availability):

I quickly looked at the code, but I didn't try running the code myself. I think this is a great resource though. It would be further strengthened by publication of traces that could be used with the code. Without data provided or simulated data provided, it is hard to test the code and evaluate it.

We appreciate this suggestion. As we noted above, all data shown in this manuscript have been included in the Source Data file in this revision. We have now also included sample data sets on the GitHub repository for each of the scripts.

Response to Reviewers

Re: Nature Communications manuscript NCOMMS-24-80678A

RNA Polymerase II is a Polar Roadblock to a Progressing DNA Fork by Kay et al.

We greatly appreciate the time and effort of all three Reviewers and sincerely thank them for their thoughtful and constructive feedback throughout the review process. We are pleased that Reviewers #2 and #3 found the revised manuscript satisfactory and had no further concerns (see their comments below). We have addressed the remaining comments from Reviewer #1 in a point-by-point response, with each comment (bold) followed by our response (not bold).

Reviewer #1 (Remarks to the Author):

The authors have done a considerable number of new experiments to improve the manuscript and have responded in detail to comments in the first review. Quite impressive!

We are grateful for the Reviewer's encouraging remarks and appreciate their recognition of the additional experiments and detailed revisions made to strengthen the manuscript.

I would recommend acceptance given attention to the below two suggestions:

(1) I have a lingering concern with respect to the interpretation of the RPA experiments presented in Supplementary Figure 7. One might expect that the presence of RPA alone (and the formation of an RPA filament on the regions of ssDNA) should act to prevent DNA rewinding. In fact, the authors show in SF7c that RPA does in fact prevent ssDNA-ssDNA interactions. If so, it would seem that one can't interpret the lack of rewinding in the presence of RPA as being due to an RNA-DNA hybrid and can't conclude that, "that hybrid formation effectively competes with RPA binding to ssDNA".

However, the control group of paused A20 polymerases, which lack the RNA to make an RNA-DNA hybrid do show that it is indeed the RNA that is required as opposed to an effect of RPA itself. This control should be pointed out in the text and used to more strongly support the conclusion.

Lastly, since there are differences between the paused and elongating polymerases, a further control (if the authors were so inclined) would be the inclusion of RNase T1 in the presence of RPA. The expectation is that one would lose the blockage of re-zipping under these conditions as is seen in the absence of RPA.

The Reviewer is correct that RPA inhibits the initial re-zipping. However, as demonstrated in the paused A20 control (Supplementary Fig. 7), in the absence of RNA, RPA will be displaced by the re-zipping fork when the force falls to 5 pN or lower, allowing the duplex DNA to rezip. When the RNA-DNA hybrid is present, however, the hybrid acts as a blockage to both the initial and final re-zipping, hindering the duplex DNA from re-zipping even when the force falls below 5 pN. To clarify this, we have revised the caption of Supplementary Fig. 7 in the revised manuscript.

As the Reviewer already noted, our control experiment with paused A20 polymerases, which lack the RNA to make an RNA-DNA hybrid, has already shown that it is indeed the RNA that is required, as opposed to an effect of RPA itself. The Reviewer suggested that an additional experiment involving RNase T1, which degrades RNA. We appreciate the reviewer's suggestion for this new control

experiment. However, we believe this additional experiment is not necessary because the presence of RNase T1 will lead to a similar re-zipping behavior as that of paused A20 polymerases and would not provide further mechanistic insights beyond what our current data set has established.

(2) I also suggest statistical analysis of the +/- RNase T1 distributions presented in Figure 1c and Supplementary Figure 4. Under which conditions are the distributions consistent with an effect?

To address the Reviewer's suggestion, we have added a statistical comparison of the (+) and (-) RNase T1 conditions from Figure 1 and Supplementary Fig. 4 using an unpaired two-sample t-test. Inclusion of this description requires a significant number of words to be added to the figure caption, exceeding the word limit on the figure caption. Therefore, we have shown this description of this analysis in the Methods section on page 18 of the manuscript.

Reviewer #2 (Remarks to the Author):

The authors adequately address my two major comments through performing the experiment in the presence of RPA, conducting additional analysis and introducing changes to the text. The revised manuscript is significantly improved. I have no further comments.

Reviewer #2 (Remarks on code availability):

There is no code, but the authors provided the raw data for the figures in Github for those who are interested in reanalyzing the data, which I think is sufficient to address any future query.

Reviewer #3 (Remarks to the Author):

The authors have done an excellent job addressing my comments. They performed additional experiments, analysis, and provided additional figures. Specifically, they now demonstrate that RNase T1 does not impact any detectable properties of the elongation complex. They also demonstrate T7 RNAP has similar polar roadblock results as RNAPII. Finally, they were not able to demonstrate fully that mechanically unzipping DNA is entirely similar to the replisome. However, they have modified the text to accurately reflect the limitations for their approach.

This paper is ready for publication. I have no further comments.